# Towards Constraint-aware Learning for Resource Allocation in NFV-enabled Networks

## Abstract

Virtual Network Embedding (VNE) is a challenging combinatorial optimization problem that refers to resource allocation associated with hard and multifaceted constraints in network function virtualization (NFV). Existing works for VNE struggle to handle such complex constraints, leading to compromised system performance and stability. In this paper, we propose a **CON**straint-**A**ware **L**earning framework for VNE, named **CONAL**, to achieve efficient constraint management. Concretely, we formulate the VNE problem as a constrained Markov decision process with violation tolerance. This modeling approach aims to improve both resource utilization and solution feasibility by precisely evaluating solution quality and the degree of constraint violation. We also propose a reachability-guided optimization with an adaptive reachability budget method that dynamically assigns budget values. This method achieves persistent zero violation to guarantee the feasibility of VNE solutions and more stable policy optimization by handling instances without any feasible solution. Furthermore, we propose a constraint-aware graph representation method to efficiently learn cross-graph relations and constrained path connectivity in VNE. Finally, extensive experimental results demonstrate the superiority of our proposed method over state-of-the-art baselines. Our code is available at https://github.com/GeminiLight/conal-vne.

## 1 Introduction

Network Function Virtualization (NFV) is a promising technique that facilitates the deployment of multiple Virtual Networks (VNs) tailored to user network demands within a shared Physical Network (PN) infrastructure (Yi et al., 2018). It is vital for domains such as cloud computing and edge computing, where dynamic and efficient resource management is essential (Zhuang et al., 2020). Virtual Network Embedding (VNE), a fundamental resource allocation problem in NFV, is critical for maintaining high Quality of Service (QoS). This process, involving the mapping of VNs to PNs, represents a significant challenge. It is an NP-hard Combinatorial Optimization Problem (COP) characterized by intricate and hard constraints (Rost & Schmid, 2020).

Traditional solutions to VNE range from exact to heuristic methods often struggle with either excessive computation times or limited performance in complex network scenarios (Zhang et al., 2018; Dehury & Sahoo, 2019; Fan et al., 2023). Recently, Reinforcement Learning (RL) has been a potential direction for VNE, which learns effective solving policies without the need of labeled datasets. UPDATE Typically, existing RL-based methods solve the VNE problem as a Markov Decision Process (MDP) (Haeri & Trajković, 2017; Yan et al., 2020; Zhang et al., 2023b). They build feature extractors with various neural networks, and then learn policies that iteratively select a physical node to place each virtual node until the solution is completed or constraints are violated. However, in the RL framework, due to the hard constraints of VNE, a persistent zero-violation is required at each decision timestep. This strict adherence to constraints often results in numerous *failure samples* where constraints are violated. For *failure samples*, existing studies (e.g., Yao et al. (2020); Zhang et al. (2022)) consider them as noisy and only train with data that violate no constraint; or others, like Yan et al. (2020) and He et al. (2023a), consider fixed penalties to them and discourage violations.

Although these RL-based algorithms (Zhang et al., 2022; He et al., 2023a; Zhang et al., 2023b) have shown efficacy, they still suffer from several significant problems in handling complex constraints of VNE. Firstly, ignoring *failure samples* makes policies prone to violating critical constraints, while employing fixed penalties in reward signals does not accurately reflect the severity of constraint violations. Thus, these methods underestimate valuable sample information and hamper the learning of constraint-aware policies, which results in low feasibility guarantees. Secondly, more seriously,

it is hard to avoid to encounter *unsolvable instances* whose feasible sets are empty in practical scenarios, due to insufficient physical resource availability or excessive virtual resource requests. It is impractical to distinguish solvable and *unsolvable instances*, since checking the instance solvability of an NP-hard problem is time-consuming. *Failure samples* caused by *unsolvable instances* further complicate the constraint learning process, and negatively impact the stability of training and policy performance. We have conducted a preliminary study to highlight the negative impact of unsolvable instances on training, please see Appendix C. Thirdly, VNE constraints are complex and multifaceted, involving cross-graph status interactions and bandwidth-constrained path connectivity assessments, which are not adequately captured by the feature extractors used in current studies. For a more comprehensive discussion on related work, please refer to Appendix A.

To address these challenges, we propose a **CON**straint-**A**ware **L**earning framework for VNE, named **CONAL**, achieving high solution feasibility guarantee and training stability. Concretely, to optimize performance while enhancing constraint satisfaction, we formulate the VNE problem as Constrained MDP (CMDP) (Altman, 2021). However, in the process of solution construction, if any constraints are violated, the process will be early terminated and lead to incomplete solutions. This challenges precise measurement of both the quality of solutions and the degree of constraint violations. Thus, we develop a violation-tolerant mapping method to ensure complete solution construction and a measurement function to precisely evaluate constraint violations. Additionally, to achieve persistent zero constraint violation required by VNE, we introduce a reachability analysis into the optimization objective. During training, due to the existence of *unsolvable instances* whose constraints are impossibly satisfied, we propose an adaptive reachability budget method to make the policy optimization more stable. It dynamically decides the violations caused by a surrogate policy as budgets based on the instance's solvability, rather than always setting budgets to zero. Furthermore, to finely perceive the complex constraints of VNE, we propose a constraint-aware graph representation method tailored for VNE. Specifically, we design a heterogeneous modeling module for cross-graph status interactions. We also devise several feasibility-consistency augmentations and utilize contrastive learning to bring representations under different views close, which enhances the sensitivity of policy towards path-bandwidth constraints. Our contributions are summarized as follows.

- We propose a new CMDP modeling approach with constraint violation tolerance for VNE, which precisely evaluate the quality of solution and the degree of constraint violation.

- We present a reachability-guided optimization objective to achieve persistent zero constraint violation required by VNE. Further to enhance the stability of policy optimization, we propose a novel adaptive reachability budget method that dynamically decides budgets.

- We propose a constraint-aware graph representation method tailored for VNE, which consists of a path-bandwidth contrast module with feasibility-consistency augmentations to perceive connectivity and a heterogeneous modeling module for cross-graph status fusion.

- We conduct extensive experiments in various network scenarios, showing the CONAL's superiority on performance, training stability, generalization, scalability and practicability.

## 2 PROBLEM DEFINITION

**System Modeling.** In real-world network systems, as illustrated in Figure 1, user network services are virtualized as VN requests that continuously seek resources from the PN. Each arrived VN request, along with the current situation of the PN, constitutes an instance $I$, and we collect all such instances with a set $\mathcal{I}$. For each instance, $I = (\mathcal{G}_v, \mathcal{G}_p) \in \mathcal{I}$, where the PN $\mathcal{G}_p$ and VN $\mathcal{G}_v$ are modeled as undirected graphs, $\mathcal{G}_p = (N_p, L_p)$ and $\mathcal{G}_v = (N_v, L_v, \omega, \varpi)$, respectively. Here, $N_p$ and $L_p$ denote the sets of physical nodes and links, indicating servers and their interconnections; $N_v$ and $L_v$ denote the sets of virtual nodes and links, representing services and their relationships; $\omega$ and $\varpi$ denote the arrival time and lifetime of VN request. We denote $\mathcal{C}(n_p)$ as the computing resource availability for the physical node $n_p \in N_p$, and $\mathcal{B}(l_p)$ as the bandwidth resource availability of the physical link $l_p \in L_p$. Besides, $\mathcal{C}(n_v)$ and $\mathcal{B}(l_v)$ denote the demands for computing resource by a virtual node $n_v \in N_v$ and bandwidth resource by a virtual link $l_v \in L_v$. Like most VNE studies (Wu et al., 2024), we focus mainly on bandwidth attributes in links, the generalizable critical bottleneck in various network scenarios, ensuring that the proposed method is universal and extensible.

**Mapping Process.** For each instance $I$, embedding a VN onto the PN can be defined as a graph mapping process, denoted $f_\mathcal{G} : \mathcal{G}_v \rightarrow \mathcal{G}_p{}'$, where $\mathcal{G}_p{}'$ is a subgraph of $\mathcal{G}_p$ that accommodates

the VN $\mathcal{G}_v$. This process comprises two sub-processes: node mapping and link mapping, where intricate hard constraints should be satisfied. Node mapping, $f_N : n_v \to n_p$, places each virtual node $n_v$ onto a physical node $n_p$, while following the one-to-one placement constraints (i.e., virtual nodes in the same VN must be placed in different physical nodes and each physical node only hosts one virtual node at most) and the computing resource availability constraints must be satisfied: $\forall n_v \in N_v, \mathcal{C}(n_v) \leq \mathcal{C}(n_p)$, where $n_p = f_N(n_v)$. Link mapping, $f_L : l_v \to p_p$, routes each virtual link through a physical path $p_p$ that connects the physical nodes hosting the two endpoints of virtual link $l_v$. This process need fulfill bandwidth resource availability constraints: $\forall l_v \in L_v, \forall l_p \in p_p, \mathcal{B}(l_v) \leq \mathcal{B}(l_p)$, where $p_p = f_L(l_v)$. If violating any of these constraints, then the VN request is rejected. Once embedded, the VN's occupied resources are released until its lifetime expires.

**VNE Objective.** To address the randomness of the network systems, same as most existing works (e.g., Zhang et al. (2022); He et al. (2023a); Zhang et al. (2023b)), we aim to learn an optimal mapping, $f_{\mathcal{G}}$, that maximizes the resource utilization of each VNE instance. This objective facilitates long-term resource utilization and request acceptance. Revenue-to-Consumption (R2C) ratio serves as a widely used metric to measure the quality of solution $E = f_{\mathcal{G}}(I)$:

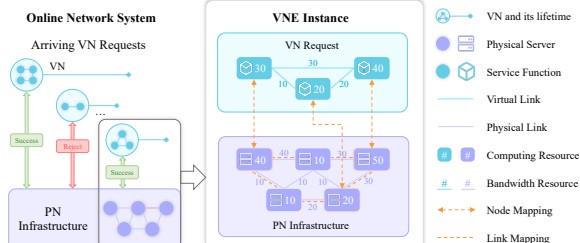

Figure 1: An brief example of VNE problem. In the network system, VN requests arrive sequentially at the infrastructure to require the resources of PN. For the VNE instance, embedding a VN to the PN consists of node and link mapping processes, while considering intricate and hard constraints.

$$\text{R2C}(E) = \varkappa \cdot (\text{REV}(E)/\text{CONS}(E)), \quad (1)$$

where $\varkappa$ is a binary variable indicating the solution's feasibility; $\varkappa = 1$ for a feasible solution and $\varkappa = 0$ otherwise. When the solution is feasible, $\text{REV}(E)$ represents the revenue from the VN, calculated as $\sum_{n_v \in N_v} \mathcal{C}(n_v) + \sum_{l_v \in l_v} \mathcal{B}(l_v)$. If $\varkappa = 1$, $\text{CONS}(E)$ denotes the resource consumption of PN, calculated as $\sum_{n_v \in N_v} \mathcal{C}(n_v) + \sum_{l_v \in L_v} (|f_L(l_v)| \times \mathcal{B}(l_v))$. Here, $|f_L(l_v)|$ quantifies the length of the physical path $p_p$ routing the virtual link $l_v$. See Appendix B for the detailed problem formulation.

## 3 METHODOLOGY

In this section, we propose the **CON**straint-**A**ware **L**earning framework to handle complex constraints of VNE, named **CONAL**, illustrated in Figure 2. Initially, we formulate the VNE problem as a violation-tolerant CMDP, which ensures the acquisition of complete solutions and precise evaluation of solution quality and constraint violation (See the green area in Figure 2). Additionally, we present a reachability-guided optimization objective to ensure persistent constraint satisfaction while avoiding the over-conservatism of policy, which enhances both the quality and feasibility of VNE solutions. Further to address the instability of policy optimization caused by instances with no feasible solution, we propose an adaptive reachability budget method to improve the robustness of training. This method dynamically decides the value of budgets rather than a fixed zero (See the pink area in Figure 2). Furthermore, regarding the feature extractor, we propose a constraint-aware graph representation method to finely perceive the complex constraints of VNE. Concretely, we construct a heterogeneous graph to model the cross-graph status between VN and PN. We also devise several augmentations that preserve solution feasibility while enhancing the model's sensitivity of bandwidth-constrained path connectivity through contrastive learning (See the orange area in Figure 2). Overall, we build the policy with the constraint-aware graph representation method and train it with an actor-critic-based RL algorithm to achieve the reachability-guided optimization objective, where the adaptive reachability budget improves the training stability. We provide the description of both CONAL's training and inference process in Algorithm 1 and 2, placed in Appendix E.

### 3.1 VIOLATION-TOLERANT CMDP FORMULATION.

To optimize resource utilization while guaranteeing solution's feasibility, we formulate solution construction of each VNE instance as a CMDP (Altman, 2021). However, VNE's hard constraints make constructing complete solutions difficult, hampering the precise assessment of solution quality

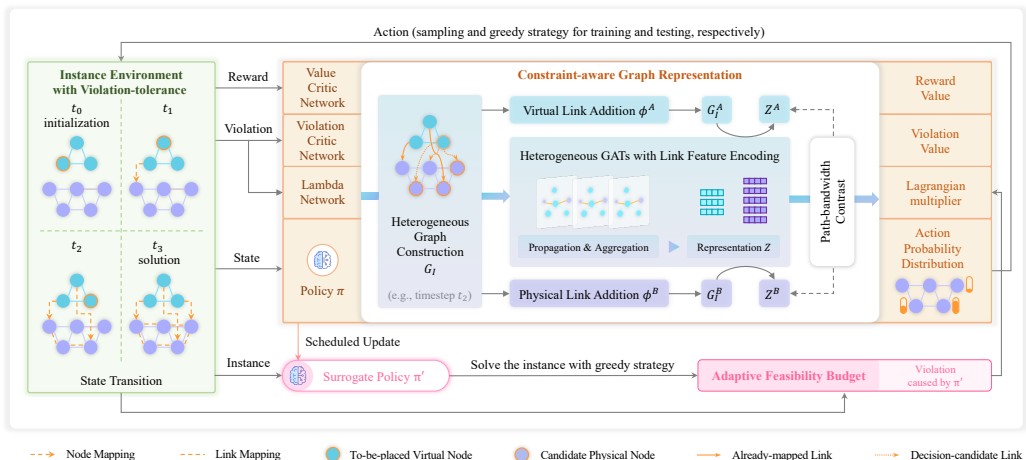

Figure 2: Overview of the proposed CONAL framework.

and the extent of constraint violations. Thus, we propose a violation-tolerant mapping method to ensure complete solution construction and customize a measurement function to evaluate violations.

**CMDP for VNE.** We consider each decision as identifying a proper physical node $n_p$ for each virtual node $n_v$ until all virtual nodes of VN are placed. Like existing works (Yan et al., 2020; Zhang et al., 2022; 2023b), due to the large combinatorial space of link mapping process, we incorporate link mapping into the state transitions, i.e., routing the *prepared incident links* $\delta'(n_v)$ of virtual node $n_v$.

**Definition 1** (Prepared incident links of virtual node). Let $\mathcal{N}_v^t$ denote the set of virtual nodes that have already been placed, and $n_v^t$ denote the to-be-placed virtual node at decision timestep $t$. We define $\delta(n_v^t)$ as the set of virtual links incident to $n_v^t$. For each virtual link $l_v \in \delta(n_v^t)$, if the link's opposite endpoint $n_v'$ is already placed, i.e., $n_v' \in \mathcal{N}_v^t$, then we include $l_v$ in a subset $\delta'(n_v^t)$. The subset $\delta'(n_v^t)$ consists of what we term as the prepared incident links of the virtual node $n_v^t$. These links are considered prepared because, upon the placement of $n_v^t$, both endpoints of each link in $\delta'(n_v^t)$ are placed, necessitating the routing of these links. See Appendix D.1 for an example.

We formulate this sequential decision process as a CMDP, $M = \langle S, A, R, H, C, P, \gamma \rangle$, where $S$ denotes the state space. Each state $s$ ($s \in S$) consists of the real-time embedding status of VN and PN. $A$ denotes the action space, i.e., the set of physical nodes. $P : S \times A \to S$ is a state transition function. For an action $a_t = n_p^t$ to host the current be-placed virtual node $n_v^t$, the environment will execute the node placing and link routing. $n_v^t$ is embedded into the $n_p^t$, and the available resources of $n_p^t$ are updated accordingly. Then, for the *prepared incident links* $\delta'(n_v^t)$ of virtual nodes $n_v^t$, we utilize the $k$-shortest path algorithm to find physical paths to route them one by one. $R : S \times A \to \mathbb{R}$ is a reward function defined as follows. If the solution is completely constructed, we return its R2C metric as the reward; for the intermediate steps, we set the reward to 0. $H : S \to \mathbb{R}$ is a violation function that measures violations of constraints. We separately consider computing and bandwidth constraint violations in node placement and link routing, denoted $H_N$ and $H_L$, respectively. We define $H(s) = \max(H_N(s), H_L(s))$ and allow it negative, which indicates keeping distance from nodes or links with insufficient resources. $C : S \to \mathbb{R}^+$ is a cost function calculated as $C(s) = \max(H(s), 0)$. $\gamma : S \times A \times S \to [0, 1]$ is a discount factor that balances immediate and future rewards. Next, we describe the violation-tolerant mapping method used in state transition, and explain the violation measurement functions tailored for $H$ and $C$.

**Violation-tolerant Mapping and Violation Measurement.** Any violations of VNE constraints result in incomplete allocation, which hinders the ability to estimate the final R2C metric and constraint violation degree of infeasible solutions. To address this issue, we consider violation tolerance for both the node mapping and link mapping processes. This tolerance enables us to execute sustainable resource allocation of VNE, despite encountering constraint violations. Concretely, at the decision timestep $t$ for placing the virtual node $n_v^t$, we generates the action probability distribution $\pi(\cdot \mid s_t)$ based on state $s_t$. If there are physical nodes with insufficient computing resources, we apply a mask vector that replaces the selection probability of these physical nodes that have insufficient node resources with 0 to avoid unnecessary constraint violations; but if all physical nodes are computing resource-insufficient, we also do not modify the action probability distribution; oth-

erwise, we do nothing. Then, an action $a_t$ (a physical node $n_p^t$) is selected from this distribution. we calculate the computing resource violations as $h_N^t = H_N(s_{t+1}) = \mathcal{C}(n_v^t) - \mathcal{C}(n_p^t)$.

Subsequently, we route all *prepared incident links* $\delta'(n_v^t)$ of virtual node $n_v^t$ sequentially. For each virtual link $l_v \in \delta'(n_v^t)$, we use the $k$-shortest path algorithm to find a set of physical paths. If there are available paths that do not violate constraints, we select the shortest one that consumes the least resources. In this case, we consider the violation as $H_L(l_v) = \max_{l_p \in f_L(l_v)}(\mathcal{B}(l_v) - \mathcal{B}(l_p))$. However, if all paths violate constraints, we calculate the extent of the constraint violation for each path. The path with the least amount of violation is then selected to route the virtual link, whose violation is calculated by $H_L(l_v) = \sum_{l_p \in f_L(l_v)}(\mathcal{B}(l_v) - \mathcal{B}(l_p))$. After $\delta'(n_v^t)$ are routed, we define the bandwidth violations as $h_L^t = H_L(s_{t+1}) = \max_{l_v \in \delta'(n_v^t)} H_L(l_v)$. During training, we always set the solution feasibility flag $\varkappa$ to 1 to measure the final R2C metric until the solution is complete.

## 3.2 Reachability-guided Optimization with Adaptive Budget

Standard CMDPs focus on optimizing discounted cumulative costs to meet long-term safety, which fails to meet the consistent satisfaction requirements of VNE in all states (Liu et al., 2021). To guarantee the solution feasibility of VNE, we consider reachability analysis into CMDP to achieve state-wise zero-violation optimization (Yu et al., 2022). This objective significantly expands the feasible set of policies and mitigates the conservativeness of the policy. Additionally, to enhance the stability of policy optimization, we propose a novel adaptive reachability budget method that dynamically decides budgets, rather than always zero budget for unsolvable instances. For policy training, we leverage the Lagrange version of the PPO algorithm (Ray et al., 2019).

**Reachability-guided Optimization Objective (REACH).** In each decision timestep $t$, based on state $s_t$, we play an action $a_t \sim \pi(\cdot|s_t)$. Then, the network system transits into the next state according to $s_{t+1} \sim P(s_t, a_t)$, and feedback a reward $r_t = R(a_t, s_t)$, a violation $h_t = H(s_{t+1})$ and a cost $c_t = C(s_{t+1})$. In each episode, we collect all sampled states and actions with a trajectory memory $\tau = (s_o, a_0, s_1, a_1, \cdots)$. Due to hard constraints of VNE, we should achieve state-wise zero violations to ensure the solution's feasibility. An intuitive way is to maximize the expected cumulative rewards $J_r(\pi) = \mathbb{E}_{\tau \sim \pi}[\sum_t \gamma^t R(s_t, a_t))]$, while restricting expected cumulative costs $J_c(\pi) = \mathbb{E}_{\tau \sim \pi}[\sum_t \gamma^t C(s_t)]$ below zero at each decision state, i.e., $\max_\pi \mathbb{E}_{\tau \sim \pi}[J_r(\pi)]$, s.t. $\mathbb{E}_{\tau \sim \pi}[J_c(\pi)] \le 0$. Existing safe RL methods learn the state and cost value functions, $V_r^\pi(s)$ and $V_c^\pi(s)$, that estimate the cumulative rewards and costs from state $s$, to solve this problem as follows.

$$\max_\pi \mathbb{E}_s\left[V_r^\pi(s)\right], \text{ s.t. } \mathbb{E}_s\left[V_c^\pi(s)\right] \le 0, \tag{2}$$

However, since the non-negativity of $C$, the optimization focuses on constraint satisfaction rather than reward maximization, which causes a highly conservative policy. Thus, we consider the Hamilton-Jacobi (HJ) reachability analysis (Bansal et al., 2017) into the CMDP to obtain a policy with the best possible performance and least violations, improving both quality and feasibility of VNE solutions. We need the following concepts to further present our optimization objective.

**Definition 2** (Feasible value function)**.** The feasible value function of a specific policy $\pi$ measures the worst long-term constraint violation, defined as $V_h^\pi(s) \triangleq \max_{t \in \mathbb{N}} H(s_t \mid s_0 = s)$. Through optimizing $\pi$, the optimal feasible state-value function can achieve the least violation of the constraints, which is defined as $V_h^\star(s) \triangleq \min_\pi \max_{t \in \mathbb{N}} H(s_t \mid s_0 = s)$.

**Definition 3** (Feasible region)**.** The feasible region $S_f$ consists of all feasible states, where at least one policy satisfies the hard constraint, defined as $S_f \triangleq \{s \in S \mid V_h^\star(s) \le 0\}$. The feasible region of a specific policy $\pi$ can be defined as $\mathcal{S}_f \triangleq \{s \in S \mid V_h^\pi(s) \le 0\}$.

The feasible value function $V_h^\pi(s)$ measures the most serious constraint violation of state $s$ on the trajectory obtained by $\pi$. Specifically, if $V_h^\pi(s) \le 0$, we have $\forall s_t, t \in \mathbb{N}, h(s_t) \le 0$, i.e., starting from the state $s$, all the states are feasible on this trajectory and the policy $\pi$ can satisfy the hard constraints. Otherwise, $V_h^\pi(s) > 0$ indicates $\pi$ may violate constraints in the future states. We call $V_h^\pi(s) \le 0$ the reachability constraint, ensuring that the $\pi$ is inside the feasible set since the state constraint could be persistently satisfied. For the VNE problem, we aim to maximize the cumulative rewards while satisfying reachability constraints to ensure persistent zero violations, formulated as,

$$\max_\pi \mathbb{E}_s\left[V_r^\pi(s) \cdot \mathbb{I}_{s \in \mathcal{S}_f} - V_h^\pi(s) \cdot \mathbb{I}_{s \notin \mathcal{S}_f}\right], \text{ s.t. } \mathbb{E}_s\left[V_h^\pi(s)\right] \le 0, \forall s \in \mathcal{S}_f, \tag{3}$$

where $\mathbb{I}$ is the indicator function. Compared to the problem (2), our reachability-guided optimization problem (3) finds the largest feasible sets, bringing less conservativeness and better performance.

To solve the problem (3), we reformulate it in the Lagrangian version as follows:

$$\min_\lambda \max_\pi \mathbb{E}_s \left[ V_r^\pi(s) \cdot \mathbb{I}_{s \in \mathcal{S}_f} - V_h^\pi(s) \cdot \mathbb{I}_{s \notin \mathcal{S}_f} + \lambda V_h^\pi(s) \cdot \mathbb{I}_{s \in \mathcal{S}_f} \right]. \tag{4}$$

**Adaptive Reachability Budget (ARB).** During training, it is hard to avoid the existence of some *unsolvable VNE instances* without any feasible solution, whose all states are infeasible, i.e., $\forall s, s \notin S_f$. For example, incoming VN requires excessive resources that surpass the resource availability of PN. Judging the solvability of an instance in an NP-hard problem is a time-consuming task, which makes it difficult to distinguish between two types of states: $s \in S_f$ and $s \notin S_f$. In this case, training with samples related to these *unsolvable instances*, due to violating the Karush-Kuhn-Tucker conditions, the Lagrange multiplier may become large and even converge to infinity.

**Proposition 1.** *During online training, if there exists an instance without any feasible solution (i.e. $H(s) > 0, \forall s \in S$), then the Lagrange multiplier can become infinite.*

For its proof see Appendix D.2. The fluctuation of $\lambda$ induces instability in policy optimization. This instability arises from significant shifts in the optimization focus, alternating between maximizing rewards and satisfying constraints. To address this challenge, we propose an adaptive reachability budget method to improve the stability of training, which determines an appropriate reachability budget for each VNE instance. To avoid the impractical determination of the solvability of each instance, we relax the zero-violation of reachability constraints with a dynamic reachability budget based on the instance solvability. Specifically, we employ a surrogate policy $\pi'$ derived from the main policy $\pi$, and synchronize its parameters at specified steps, i.e., $\pi' \leftarrow \pi$. During the training process, both the main policy $\pi$ and the surrogate policy $\pi'$ attempt to solve the same incoming instance $I$. $\pi$ uses the sampling decoding strategy for exploration to generate the trajectory $\tau \sim \pi$, while $\pi'$ employs the greedy decoding strategy for prioritizing constraint satisfaction to produce the trajectory $\tau' \sim \pi'$. The max cost in $\tau'$ caused by the surrogate policy $\pi'$ is considered for estimating the reachability budget $D_h^{\pi'}(s)$ for all states $s \in \tau$ sampled by policy $\pi$, formulated as follows:

$$\forall s \in \tau, D_h^{\pi'}(s) = \max_{s' \in \tau'} C(s'). \tag{5}$$

During the training process, we update the surrogate policy $\pi' \leftarrow \pi$ over multiple iterations. With the reachability guidance of $\pi'$, the main policy $\pi$ gradually improves constraint-aware decision-making, enhancing the stability of training. Through this iterative learning, both policies $\pi$ and $\pi'$ achieve better constraint satisfaction while maintaining the exploration for better solutions. Finally, we obtained the refined Lagrangian objective with the adaptive reachability budget as follows:

$$\min_\lambda \max_\pi \mathbb{E}_s \left[ V_r^\pi(s) - \lambda \left( V_h^\pi(s) - D_h^{\pi'}(s) \right) \right]. \tag{6}$$

Here, considering the varying extent of violation in different states, we introduce a neural lambda network $\lambda = \Lambda(s)$ to dynamically adjust Lagrangian multipliers during training, similar to Ma et al. (2021). To optimize the policy, we leverage the actor-critic framework with Proximal Policy Optimization (PPO) as the training algorithm (Ray et al., 2019), similar to works (Yu et al., 2022; Ma et al., 2021). See Appendix D.5 for the details of this training method. Next, we will introduce our proposed constraint-aware graph networks used as the feature encoder of policy.

### 3.3 CONSTRAINT-AWARE GRAPH REPRESENTATION

The VNE processing is governed by complex and multifaceted constraints, presenting challenges in representation learning. These include the interaction of cross-graph status and the assessment of bandwidth-constrained path connectivity. To address this issue, we propose a constraint-aware graph representation method with a heterogeneous modeling module for cross-graph status fusion and a contrastive learning-based module to enhance path connectivity awareness. This method efficiently perceives the complex constraints of VNE, providing a higher feasibility guarantee of solutions.

**Heterogeneous Modeling (HM).** Instead of separate feature extraction of VN $G_v$ and PN $G_p$, we integrate them into a heterogeneous graph $G_I$ by introducing several hypothetical cross-graph links.

$G_I$ comprises two distinct node types: virtual and physical nodes, and we denote their attributes as $X_v^n$ and $X_p^n$. These node attributes include computing resource demands or availability, link counts, and the aggregated bandwidth characteristics (maximum, minimum, and average) of adjacent links. Similarly, there are two link types: virtual and physical links, whose attributes are link bandwidth demands or availability. We denote these link attributes as $X_v^l$ and $X_p^l$, respectively. Additionally, we introduce specialized heterogeneous links to capture the current embedding state: already-mapped links, which connect virtual nodes to their hosting physical nodes, and imaginary decision links, which connect the current yet-to-be-decided virtual node with all *potential physical nodes*. Here, *potential physical node* refers to one that hosts no virtual nodes and has enough computing resources available. We group these heterogeneous links into sets $L_{v,p,m}$ for already-mapped links and $L_{v,p,d}$ for decision-candidate links. We uniformly set these two types of links' attributes to 1, denoted as $X_{v,p,m}^l$ and $X_{v,p,d}^l$, respectively. To encode this graph's topological and attribute information, we enhanced widely-used graph attention networks (GAT) (Veličković et al., 2018) by integrating heterogeneous link fusion and link attribute encoding in the propagation process. See Appendix D.3 for its details. Inputting the heterogeneous graph's features into this network, we obtain the final node representations $Z = \{Z_v, Z_p\}$, where $Z_v$ and $Z_p$ denote physical and virtual node representations.

**Path-bandwidth Contrast (PC).** Bandwidth constraints of VNE significantly impact solution feasibility, particularly in the context of path routing complexity. At each decision timestep, we need to carefully select a physical node $n_p^t$ for placing the current virtual node $n_v^t$. This selection is dominated by ensuring that feasible connective paths exist to all other physical nodes hosting the virtual node's neighbors. Here, the feasibility of the path is dominated by the bandwidth availability of physical links to support the bandwidth requirements of all *prepared incident links* $\delta'(n_v^t)$. GNNs build up on the propagation mechanism along links to increase awareness of the topology information. However, not all physical links contribute positively to this awareness; some with insufficient bandwidths may even introduce noise into node representations. This emphasizes the necessity to integrate bandwidth constraint awareness within GNNs to perceive the path feasibility.

To address this challenge, we propose a novel path-bandwidth contrast method to enhance bandwidth constraint awareness through contrastive learning, whose core idea is creating augmented views with feasibility-consistency augmentations and making node representations in these views close.

**Definition 4** (Feasibility-consistency Augmentations). Let $\Phi$ denote a set of augmentation methods and $\mathcal{F}$ denote the function indicating the feasibility of solutions. Given any VNE instance $I$ and a solution $E = f_G(I)$, we have $\mathcal{F}(E) = \mathcal{F}(\phi(E)), \forall I \in \mathcal{I}, \phi \in \Phi$.

These augmentations generate multiple views of the original heterogeneous graph, which maintain the same feasibility semantics before and after their application. Following this principle, we develop several augmentation methods by modifying the topology of either VN or PN without impacting solution feasibility, which are described as follows. (a) *Physical Link Addition* $\phi^A$. We add a specific number $\epsilon \cdot |N^p|$ of physical links in PN, whose bandwidth resources are equal to the difference between the smallest requirements among all virtual links and 1, i.e., $\min_{l_v \in L_v} \mathcal{B}(l_v) - 1$. (b) *Virtual Link Addition* $\phi^B$. We add a specific number $\epsilon \cdot |N^v|$ of virtual links that require a zero bandwidth resource to enhance the complexity and connectivity of the VN. Here, $\epsilon$ is an augment ratio that determines the proportion of links to be added based on the number of nodes in the network.

After applying these augmentations, we create two new views $G_I^A = \phi_A(G_I)$ and $G_I^B = \phi^B(G_I)$, which have same feasibility semantics of VNE instance $I$. Using our heterogeneous graph network, we extract node representations under views $G_I^A$ and $G_I^B$, denoted as $Z^A$ and $Z^B$, respectively. Subsequently, we utilize contrastive learning to enhance the proximity of node representations under the augmented views $G_I^A$ and $G_I^B$. This necessitates that the model precisely discerns the noisy implications of those links with less bandwidth that play no impact on solution feasibility. Through this method, we aim to enhance the model's sensitivity towards link bandwidth, effectively mitigate the influence of irrelevant links in the GNN propagation process, and bolster its overall awareness of bandwidth constraints. In this work, we adopt the Barlow Twins method (Zbontar et al., 2021) for its simplicity and effectiveness, which circumvents negative sample selection and maintains the original network architecture. Given the embeddings under two augmented views, $H^a$ and $H^b$, we use this contrastive loss $L_{CL}$ to reduce redundancy between embedding components by aligning their cross-correlation matrix with the identity matrix. This unsupervised loss can be seamlessly integrated into the training process of RL. See Appendix D.4 for the details of Barlow Twins method.

Table 1: Results in overall evaluation and ablation study. Each value consists of the mean and standard error.

| | VN_RAC ↑ | LT_R2C ↑ | LT_REV ($\times 10^7$) ↑ | C_VIO ($\times 10^3$) ↓ | AVG_ST ($\times 10^{-1}$ s) ↓ |
|---|---|---|---|---|---|
| NRM-VNE | $0.675 \pm 0.011$ | $0.461 \pm 0.003$ | $7.649 \pm 0.089$ | - | $\mathbf{1.285 \pm 0.042}$ |
| GRC-VNE | $0.694 \pm 0.020$ | $0.468 \pm 0.004$ | $7.888 \pm 0.081$ | - | $\underline{2.737 \pm 0.056}$ |
| NEA-VNE | $0.732 \pm 0.017$ | $\underline{0.558 \pm 0.007}$ | $8.635 \pm 0.183$ | - | $4.471 \pm 0.656$ |
| GA-VNE | $0.735 \pm 0.043$ | $0.477 \pm 0.007$ | $8.355 \pm 0.082$ | - | $47.462 \pm 1.117$ |
| PSO-VNE | $0.723 \pm 0.025$ | $0.456 \pm 0.004$ | $7.854 \pm 0.060$ | - | $51.955 \pm 3.512$ |
| MCTS-VNE | $0.700 \pm 0.085$ | $0.477 \pm 0.006$ | $7.809 \pm 0.394$ | - | $15.512 \pm 8.096$ |
| PG-CNN | $0.682 \pm 0.020$ | $0.487 \pm 0.004$ | $7.523 \pm 0.156$ | - | $3.906 \pm 0.057$ |
| DDPG-ATT | $0.707 \pm 0.021$ | $0.469 \pm 0.003$ | $7.961 \pm 0.091$ | - | $2.991 \pm 0.054$ |
| A3C-GCN | $0.743 \pm 0.019$ | $0.540 \pm 0.006$ | $8.814 \pm 0.223$ | - | $3.585 \pm 0.200$ |
| GAL-VNE | $\underline{0.776 \pm 0.014}$ | $0.495 \pm 0.003$ | $9.267 \pm 0.162$ | - | $6.881 \pm 0.785$ |
| CONAL$_{\text{w/o HM}}$ | $0.804 \pm 0.044$ | $0.584 \pm 0.004$ | $9.597 \pm 0.107$ | $3.410 \pm 0.080$ | $3.754 \pm 0.124$ |
| CONAL$_{\text{w/o PC}}$ | $0.735 \pm 0.036$ | $0.573 \pm 0.002$ | $8.407 \pm 0.128$ | $5.960 \pm 0.068$ | $4.117 \pm 0.273$ |
| CONAL$_{\text{w/o HM \& PC}}$ | $0.789 \pm 0.058$ | $0.585 \pm 0.004$ | $9.446 \pm 0.084$ | $4.053 \pm 0.072$ | $3.909 \pm 0.104$ |
| CONAL$_{\text{w/o REACH}}$ | $0.792 \pm 0.049$ | $0.611 \pm 0.007$ | $9.607 \pm 0.073$ | $3.954 \pm 0.045$ | $4.052 \pm 0.069$ |
| CONAL$_{\text{w/o ARB}}$ | $0.806 \pm 0.034$ | $0.596 \pm 0.003$ | $9.639 \pm 0.065$ | $3.656 \pm 0.087$ | $4.074 \pm 0.093$ |
| **CONAL** | $\mathbf{0.813 \pm 0.042}$ | $\mathbf{0.614 \pm 0.006}$ | $\mathbf{9.842 \pm 0.091}$ | $\mathbf{2.773 \pm 0.083}$ | $4.180 \pm 0.104$ |

## 3.4 Computational Complexity Analysis

Note that CONAL solely uses its surrogate policy and path-bandwidth contrast module for policy optimization during training. During inference, CONAL has a computational complexity of $O\left(|N_v| \cdot K \cdot \left(|L_p|d + |N_p + N_v|d^2\right)\right)$, while the complexities of baseline methods based on RL and GNNs are $O\left(|N_v| \cdot K \cdot \left(|L_p|d + |N_p|d^2\right)\right)$. See Appendix D.6 for detailed explanations. While CONAL slightly increases the complexity compared to existing RL and GNN-based methods due to its heterogeneous modeling approach, it achieves significant performance improvements.

## 4 Experiments

In this section, we describe the experimental setup. To evaluate the effectiveness of CONAL, we compare it with several variations and state-of-the-art baselines in various network scenarios.

### 4.1 Experimental Settings

**Simulation Configurations.** Similar to most previous works (Geng et al., 2023; Yan et al., 2020), we evaluate the proposed framework in the simulation benchmarks that mimic various network systems. We adopt a Waxman topology with 100 nodes and nearly 500 links (Waxman, 1988) as the physical network, named **WX100**. Computing resources of physical nodes and bandwidth resources of physical links are uniformly distributed within the range of [50, 100] units. In default settings, for each simulation run, we create 1000 VN with varying sizes from 2 to 10. The computing resource demands of the nodes and the bandwidth requirements of the links within each VN are uniformly distributed within the range of [0, 20] and [0, 50] units, respectively. The virtual nodes in each VN are randomly interconnected with a probability of 50%. The lifetime of each VN follows an exponential distribution with an average of 500 time units. The arrival of these VNs follows a Poisson process with an average rate $\eta$, where $\eta$ denotes the average arrived VN count per unit of time. In the subsequent experiments, we manipulate the distribution settings of VNs and change the PN topologies to simulate various network systems.

**Implementation Settings.** We describe the details of CONAL implementations, simulation for training and testing, and computer resources in Appendix F.1. Notably, in scenarios where the PN topology remains unchanged, we employ the pre-trained models developed under default settings to investigate the adaptability and generalization across diverse conditions in network systems.

**Compared Baselines.** We compare CONAL with both heuristic and RL-based methods. The heuristic baselines includes node ranking-based methods (i.e., NRM-VNE (Zhang et al., 2018), GRC-VNE (Gong et al., 2014), NEA-VNE (Fan et al., 2023)) and meta-heuristics (i.e., GA-VNE (Zhang et al., 2019), PSO-VNE (Jiang & Zhang, 2021)). The learning-based baselines are PG-CNN (Ma et al., 2023), DDPG-ATT (He et al., 2023a), A3C-GCN (Zhang et al., 2023b), and GAL-VNE (Geng et al., 2023). See Appendix F.2 for their descriptions.

**Evaluation Metrics.** Following most previous research (Fischer et al., 2013; Yan et al., 2020), we evaluate the effectiveness of VNE algorithms with widely used key performance metrics: VN Acceptance Rate (**VN_ACR**); Long-Term REVenue (**LT_REV**); Long-Term Revenue-to-Consumption

ratio (**LT_R2C**). We also consider **AV**era**G**e Solving Time (**AVG_ST**) as an additional metric to measure the computational efficiency, due to the real-time demands of online network systems. Additionally, for CONAL and its variations, we employ Constraint VIOlation (**C_VIO**) to measure the degree of constraint satisfaction. See Appendix F.4 for their definitions.

## 4.2 RESULTS AND ANALYSIS

**Overall Evaluation.** The results of VNE algorithms under default settings are shown in Table 1. We observe that CONAL outperforms baselines across three key performance metrics. This is due to CONAL's ability to effectively perceive and handle the complex constraints of VNE. Among baselines, NEA-VNE, GA-VNE, and GAL-VNE are the best node-ranking-based heuristics, meta-heuristics, and RL-based methods. However, GA-VNE which relies on extensive search in the solution space has a longer running time, while GAL-VNE exhibits lower R2C metrics. Compared to NEA-VNE, GA-VNE and GAL-VNE, CONAL achieves improvements of (10.04%, 11.07%, 13.99%), (10.61%, 28.72%, 17.80%) and (4.77%, 24.04%, 6.21%) in term of (VN_ACR, LT_R2C, and LT_REV), respectively. We also observe that CONAL outperforms all RL-based baselines across key performance metrics, demonstrating the superiority of its modeling, optimization, and representation methods. Although CONAL's running time is not the lowest, it remains competitive, comparable to NEA-VNE and A3C-GCN, and outperforms other baselines such as GA-VNE and MCTS-VNE. These results underscore the effectiveness of CONAL for VNE in providing high-quality and feasible solutions to improve resource utilization and request acceptance.

**Ablation Study.** We design several variations to manifest the efficacy of each proposed component: CONAL$_{\text{w/o HM}}$, CONAL$_{\text{w/o PC}}$, CONAL$_{\text{w/o HM \& PC}}$, CONAL$_{\text{w/o REACH}}$, and CONAL$_{\text{w/o ARB}}$. See Appendix F.3 for their descriptions. The results of CONAL and its variations are presented in Table 1. Compared to the first three variants, CONAL achieves superior results across various performance metrics. This indicates that our constraint-aware graph representation method enhances the constraint awareness of policy. Notably, CONAL$_{\text{w/o PC}}$ shows the most significant performance declines, even worse than CONAL$_{\text{w/o HM \& PC}}$. This may be due to heterogeneous graph modeling methods increasing the link complexity of graph, which highlights the necessity for improving bandwidth constraint sensitivity. Furthermore, CONAL$_{\text{w/o REACH}}$ and CONAL$_{\text{w/o ARB}}$ also decrease performance, which demonstrates that our optimization method facilitates both resource utilization and constraint satisfaction. This study shows that each component of CONAL contributes to its overall performance, enhancing the its ability to handle and perceive the complex constraints of VNE.

**Training Stability Study.** To study the training stability of CONAL, we first compare the learning curves of CONAL and CONAL$_{\text{w/o ARB}}$, followed by an analysis of how different training conditions impact testing performance. (A) *Learn curves analysis*. We compare the learning curves of CONAL and CONAL$_{\text{w/o ARB}}$, depicted in Figure 3. We average the episode returns of 1000 VNs in one simulation as the metric, due to inherent differences of return resulting from VN size. We observe that while the performance of CONAL$_{\text{w/o ARB}}$ fluctuates during the training process, CONAL exhibits greater stability. This stability is attributed to our adaptive reachability budget method, which effectively addresses unsolvable instances during

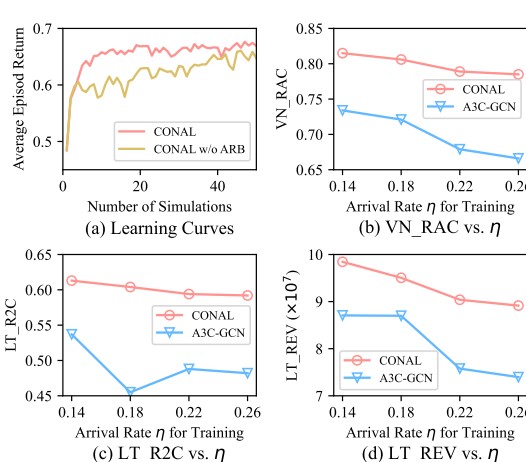

Figure 3: Results in Training Stability Study.

training, thus enhancing overall training stability. (B) *Training Conditions vs. Testing Performance Analysis*. Similar to our preliminary study, we further explore the impact of training conditions on CONAL performance. We train CONAL and A3C-GCN under different arrival rates $\lambda$ of VN requests from 0.14 to 0.26, as higher arrival rates increase the frequency of unsolvable instances. After training, we evaluate the models using a fixed arrival rate of $\lambda = 0.14$ with a random seed of 0 as the benchmark. The results are shown in Figures 3(b)(c)(d). We observe that as the arrival rate $\lambda$ of training conditions increases, CONAL maintains a more stable testing performance while A3C-GCN shows an obvious decline. This analysis shows that CONAL is more stable in learning a high-quality policy by efficiently handling unsolvable instances and perceiving complex constraints.

**Generalizability Study.** In practical network systems, fluctuations in traffic patterns and resource demands are inevitable due to varying service requirements and dynamic user behaviors. To study the generalizability, we test the trained CONAL model in various network conditions via the following experiments. See Appendix G.1 for details. (A) *Request Frequency Sensitivity Study.* We assess the sensitivity of CONAL to varying arrival rates of VN requests by adjusting the average request frequency $\eta$. As illustrated in Figure 5, CONAL consistently outperforms the baseline algorithms across all tested $\eta$ values, demonstrating its superior adaptability to changes in network traffic. This analysis highlights CONAL's effectiveness in handling network scenarios with fluctuating request rates, ensuring stable performance even as competition for resources increases. (B) *Dynamic Request Distribution Study.* To simulate more realistic network conditions, we evaluate CONAL's performance under varying VN request distributions by modifying resource demands and node sizes in different stages. As shown in Figure 6, CONAL demonstrates strong adaptability across all stages, outperforming the baseline algorithms even as the complexity of requests increases. This study shows CONAL's ability to generalize effectively in dynamic networks with shifting requirements.

**Scalability Analysis.** To assess the scalability of CONAL, we explore its performance in large-scale network systems and its time consumption to adapt to varying network sizes. See Appendix G.2 for details. In summary: (A) *Large-scale Network Validation.* We evaluate CONAL on a Waxman topology with 500 nodes and about 1300 links mimicking a large-scale cloud cluster. The results are shown in Figure 8. We observe that CONAL outperforms the baselines in most metrics in such a larger topology. This demonstrates the efficiency of CONAL in large-scale network scenarios. Results demonstrate that CONAL consistently outperforms all baseline models, even in this large-scale network system scenario. (B) *Solving Time Scale Analysis.* To investigate solving time scalability of CONAL, we increase the size of physical network from 200 to 1,000 nodes to simulate network systems of varying scales. The results, depicted in Figure 7, show that even at larger network scales, CONAL maintains efficient solving times while delivering excellent performance. This efficiency make CONAL a viable solution for real-time decision-making in large-scale network systems.

**Real-world Network Topology Validation.** To verify the effectiveness of CONAL in real-world network systems, we conduct experiments on two well-known networks (Yan et al., 2020; He et al., 2023a): GEANT, a 40-node academic research network with 64 edges, and BRAIN, a 161-node high-speed data network with 166 edges. The results shown in Table 3 reveals that CONAL outperforms all baselines across both network systems on performance metrics. This validation shows the effectiveness of CONAL in practical systems and various topologies. See Appendix G.2 for details. **UPDATE**

**Hyperparameter Sensitivity Study.** We explore the impact of the following two key hyperparameters on performance. (A) *Update interval $\mu$ of surrogate policy*. The results are shown in Figure 9. We observe that extremely frequent updates potentially lead to instability and divergence in the learning process. Additionally, too slow updates do not offer significant further benefits and may even increase computational overhead. (B) *Augment ratio $\epsilon$ used in the path-bandwidth contrast module*. The results shown in Figure 10 reveal that a reasonable augment ratio $\epsilon$ enhances the model's sensitivity to bandwidth constraints. However, excessively high values of $\epsilon$ yield minimal improvements or may even harm performance. See Appendix G.4 for details.

## 5 CONCLUSION

In this paper, we proposed the CONAL for VNE to enhance constraint management and training robustness, which is critical to the performance and reliability of network systems. Specifically, we formulated the VNE problem as a violation-tolerant CMDP to optimize both the quality and feasibility of solutions. This method allows us to always obtain complete solutions to precisely evaluate the quality of the solution. Additionally, we presented a reachability-guided optimization objective with an adaptive feasibility budget method to ensure persistent constraint satisfaction while alleviating the conservativeness of policy. This approach also address the instability of policy optimization caused by unsolvable instances. Furthermore, to finely perceive the complex constraints of VNE, we proposed a constraint-aware graph representation method, which consists of a heterogeneous modeling module for indicating cross-graph relations and a path-bandwidth contrast module that enhances the sensitivity to bandwidth constraints. Finally, we conducted extensive experiments to verify the effectiveness of our proposed methods. In the future, we plan to tackle additional constraints in specific networking scenarios, e.g., latency-aware edge computing and energy-efficient green computing. This will require specialized design efforts to address scenario-specific challenges effectively.

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

CONTENTS OF APPENDIX

# A    RELATED WORK

In this section, we discuss related work on VNE algorithms, RL for COPs, and safe RL methods.

**VNE Algorithms.** Resource management is a critical research direction in NFV, including tasks as Scaling (Fei et al., 2018; 2020) and scheduling (Zhang et al., 2017). Among these, VNE plays a key role in resource allocation. To solve this challenging and significant problem, many approaches **NEW** have been designed for VNE, which can be classified as exact, heuristics, and learning-based methods. Initially, exact algorithms formulate the VNE problem as integer linear programming (Shahriar et al., 2018) or mixed integer linear programming (Chowdhury et al., 2009), and then solve them with exact solvers. However, this is impractical due to extensive computations and time consumption. Thus, numerous heuristic algorithms have been proposed to offer solutions within an acceptable time, such as node ranking strategies (Zhang et al., 2018; Gong et al., 2014; Fan et al., 2023) (Jin et al., 2020), meta-heuristics (Dehury & Sahoo, 2019; Zhang et al., 2019; Jiang & Zhang, 2021), etc. **NEW** However, they heavily rely on manual heuristic design and merely for specific scenarios. Recently, reinforcement learning has emerged as a promising solution for VNE and many RL-based VNE algorithms have been proposed (Haeri & Trajković, 2017; Wang et al., 2021; Zhang et al., 2022; He et al., 2023a; Zhang et al., 2023b; Geng et al., 2023) (Xiao et al., 2019). In general, they model the **NEW** solution construction process of each VNE instance as an MDP, but do not consider the fine-grained constraint violations. Then, they leverage existing network networks (e.g., convolutional neural network, GNN, etc.) to extract features from PN and VN, separately. Finally, they optimize the model with different RL methods (e.g., asynchronous advantage actor-critic, PPO, etc.). In particular, Gu et al. (2020) proposed a model-assisted DRL framework that leverages heuristic solutions to guide the training process, reducing reliance on the agent's blind exploration of actions. However, they struggle to handle such complex constraints of VNE thereby compromising performance. Zeng et al. (2024) introduced the SafeDRL algorithm that corrects constraint violations using high-quality feasible solutions through expert intervention. But this reliance on external corrections ignore the aspects of policy-level constraints awareness, limiting its adaptability and performance. To address these challenges, we explore to learn a constraint-aware VNE policy by innovating existing MDP modeling, representation learning, and policy optimization **NEW**

**RL for COPs.** The application of RL to solve COPs has emerged as a hot topic in decision-making, which aims to learn efficient solving strategies from data (Bengio et al., 2021). Many efforts have been directed toward classic COPs such as routing (Zhou et al., 2023; Ye et al., 2023), scheduling (Zhang et al., 2023a; 2020), bin packing (Pan et al., 2023; Zhao et al., 2021), etc. These approaches can be broadly categorized into two types based on their solving processes: construction and improvement. While improvement methods start with an initial solution and use an RL policy to iteratively refine it, construction methods build a solution incrementally from scratch (Mazyavkina et al., 2021). Construction methods typically employ RL to guide the sequential selection to form a complete solution. Given the real-time requirements of practical network systems, most existing RL-based VNE algorithms are designed as construction methods to provide solutions within an acceptable time (Zhang et al., 2022; He et al., 2023a; Zhang et al., 2023b). In contrast to these classic COPs, VNE presents unique complexities in representation learning due to its multifaceted and hard constraints, such as the interaction of cross-graph status and the assessment of bandwidth-constrained path connectivity. Furthermore, the existence of unsolvable instances in the training process can compromise robustness, potentially causing ineffective policies.

**Safe RL Methods.** Safe RL aims to maximize the expected rewards while ensuring safety constraints are not violated (Gu et al., 2022). Early efforts in safe RL focus on keeping cumulative constraint violations below a fixed cost budget (Achiam et al., 2017; Ray et al., 2019; Yang et al., 2022). To address the stricter constraint requirements of practical applications, recent works have proposed achieving state-wise safety, which ensures the satisfaction of instantaneous constraints at each decision timestep (Zhao et al., 2023; 2022; He et al., 2023b; Yu et al., 2022). One promising approach involves incorporating reachability analysis into the CMDP framework (Yu et al., 2022). This method employs a reachability function to assess the state feasibility, which significantly expands the feasible set of policy, and mitigates the conservativeness of the policy. However, modeling VNE as a reachability-guided CMDP presents challenges. Concretely, In the process of solution construction, if any constraints are violated, the process will be early terminated and lead to incomplete solutions. This hinders precise measurement of both the quality of solutions and the degree of constraint violations. Additionally, these methods typically assume a non-empty feasible set. But

for VNE, where it is hard to avoid facing unsolvable instances without any feasible solution, these approaches often confront the challenges of unstable policy optimization, since the constraints of these instances are impossibly satisfied.

# B   PROBLEM FORMULATION

In this section, we provide the mathematical programming formulation of the VNE problem.

## B.1   OPTIMIZATION OBJECTIVES

The main goal of VNE is to make full use of the physical network resources to improve the revenue of ISPs while satisfying the service requests of users as much as possible. To address the stochastic nature of online networking, we and existing studies (Wang et al., 2021; He et al., 2023a; Zhang et al., 2023b), aim to minimize the embedding cost of each arriving VN request onto the physical network. This way enhances resource utilization and improves the VN request acceptance rate. To evaluate the quality of solution $E = f_G(I)$, we employ the widely used indicator, Revenue-to-Consumption ratio (R2C), defined as follows:

$$\text{R2C}\,(E) = (\varkappa \cdot \text{REV}\,(E))\,/\text{CONS}\,(E)\,. \tag{7}$$

Here, $\varkappa$ is a binary variable representing the feasibility of a solution: $\varkappa = 1$ if the solution $E$ for the instance $I$ is accepted, and $\varkappa = 0$ otherwise. $\text{REV}(E)$ denotes the revenue generated by the VN request $G_v$ and $\text{CONS}(E)$ denotes the embedding consumption, which are computed as follows:

$$\text{REV}(E) = \sum_{n_v \in N_v} \mathcal{C}(n_v) + \sum_{l^v \in L_v} \mathcal{B}(l^v), \tag{8}$$

$$\text{CONS}(E) = \sum_{n_v \in N_v} \mathcal{C}(n_v) + \sum_{l^v \in L_v} |f_L(l_v)|\mathcal{B}(l^v), \tag{9}$$

where $|f_L(l_v)|$ denotes the hop count of the physical path $p_p = f_L(l_v)$ routing the virtual link $l_v$.

## B.2   CONSTRAINT CONDITIONS

The process of embedding a VN request $G_v$ onto the physical network is represented by a mapping function $f_G : G_v \rightarrow G_p$. In this process, we need to decide two types of boolean variables: (1) $x_i^m = 1$ if virtual node $n_v^m$ is placed in physical node $n_p^i$, and 0 otherwise; (2) $y_{i,j}^{m,w} = 1$ if virtual link $l_{m,w}^v = (n_v^m, n_v^w)$ traverses physical link $l_{i,j}^p = (n_p^i, n_p^j)$, and 0 otherwise. Here, $m$ and $w$ are identifiers for virtual nodes, while $i$ and $j$ are identifiers for physical nodes. A VN request is successfully embedded if a feasible mapping solution is found, satisfying the following constraints:

$$\sum_{n_p^i \in n_p} x_i^m = 1, \forall n_v^m \in n_v, \tag{10}$$

$$\sum_{n_v^m \in N_v} x_i^m \leq 1, \forall n_p^i \in N_p, \tag{11}$$

$$x_i^m \mathcal{C}(n_v^m) \leq \mathcal{C}(n_p^i), \forall n_v^m \in N_v, n_p^i \in N_p, \tag{12}$$

$$\sum_{n_p^i \in \Omega(n_p^k)} y_{i,k}^{m,w} - \sum_{n_p^j \in \Omega(n_p^k)} y_{k,j}^{m,w} = x_k^m - x_k^w, \forall l_{m,w}^v \in L_v, n_v^k \in N_p, \tag{13}$$

$$y_{i,j}^{m,w} + y_{j,w}^{m,w} \leq 1, \forall l_{m,w}^v \in L_v, l_{i,j}^p \in L_p, \tag{14}$$

$$\sum_{l_{m,w}^v \in L_v} (y_{i,j}^{m,w} + y_{j,i}^{m,w})\mathcal{B}(l_{m,w}^v) \leq \mathcal{B}((l_{i,j}^p)), \forall(l_{i,j}^p) \in L_p. \tag{15}$$

Here, $\Omega(n_p^k)$ denotes the neighbors of $n_p^k$. Constraint (10) ensures that every virtual node is mapped to one and only one physical node. Conversely, constraint (11 )limits each physical node to hosting

at most one virtual node, thus enforcing a unique mapping relationship. Constraint (12) verifies that virtual nodes are allocated to physical nodes with adequate resources. Following the principle of flow conservation, constraint 13 guarantees that each virtual link $(n_v^m, n_v^w)$ is routed along a physical path from $n_p^i$ (the physical node where $n_v^m$ is placed) to $n_p^j$ (the physical node where $n_v^w$ is placed). Constraint (14) eliminates the possibility of routing loops, thereby ensuring that virtual links are routed acyclically. Lastly, constraint 15 ensures that the bandwidth usage on each physical link remains within its available capacity. Overall, constraints (10,11,12) enforce the one-to-one placement and computing resource availability required in the node mapping process. And constraints (13,14,15) ensure the path connectivity and bandwidth resource availability asked in the link mapping process.

## C  PRELIMINARY STUDY

We have conducted a preliminary study to highlight the motivation to handle failure samples and unsolvable instances. In this study, we trained a representative baseline model, A3C-GCN Zhang et al. (2023b), which shares a solution construction paradigm similar to our approach and uses a penalty mechanism to handle all failure samples. The training was conducted with various arrival rates for VN requests because it is evident that increasing the arrival rate of VN requests leads to a higher frequency of unsolvable instances. After training, we tested the models using a fixed arrival rate $\lambda = 0.14$ (the same as in Section 4.1: Experimental Settings) with a seed of 0 as the benchmark.

| $\lambda$ for Training | VN_RAC $\uparrow$ | LT_R2C $\uparrow$ | LT_REV ($\times 10^7$) $\uparrow$ |
|---|---|---|---|
| 0.14 | 0.734 | 0.537 | 8.707 |
| 0.18 | 0.721 | 0.455 | 8.699 |
| 0.22 | 0.679 | 0.488 | 7.578 |
| 0.26 | 0.666 | 0.482 | 7.397 |

Table 2: The testing performance of baseline A3C-GCN trained under various $\lambda$ values.

The results are shown in Figure 2. We observe that as the arrival rate of VN requests increases, the A3C-GCN model trained under higher arrival rates exhibits worse performance. Due to the increased proportion of unsolvable instances, the caused failure samples interfere more strongly with the center of policy optimization, making it difficult to learn a high-quality solution strategy. This indicates that this method struggles to effectively handle unsolvable instances during training, which negatively impacts the optimization robustness and overall performance.

## D  MODEL DETAILS

In this section, we present the key concepts, theoretical foundations, explanation of CONAL's components, and descriptions of both the training and inference processes for CONAL.

### D.1  ILLUSTRATIVE EXPLANATION OF PREPARED INCIDENT LINKS

To make reader clearly understand the prepared incident links, we provide a brief illustrative example of this concept in Figure 4. Considering the third decision timestep, after mapping virtual nodes $n_v^1$ and $n_v^2$ onto physical nodes $n_p^1$ and $n_p^2$, we aim to find a physical node to host the to-be-placed virtual node $n_v^3$. When attempting to place $n_v^3$ onto $n_p^3$, we need to consider the routing of virtual links $(n_v^1, n_v^3)$ and $(n_v^2, n_v^3)$ whose endpoints are now both placed. In this example, $(n_v^1, n_v^3)$ and $(n_v^2, n_v^3)$ are called prepared incident links. For current decision timestep, we need to route all of them with connective physical paths while ensuring resource constraints are met.

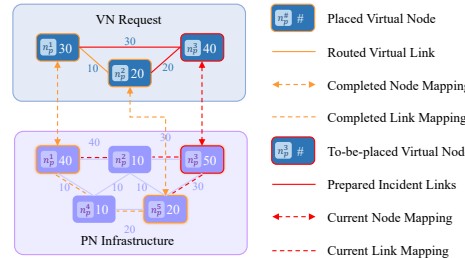

Figure 4: An illustrative example of prepared incident links.

## D.2 PROOF OF LAGRANGE MULTIPLIER CONVERGENCE

**Proposition 1.** *During online training, if there exists an instance without any feasible solution (i.e. $H(s) > 0, \forall s \in S$), then the Lagrange multiplier can become infinite.*

*Proof.* According to the optimality condition of Karush-Kuhn-Tucker (KKT), any optimal solution to the constrained optimization problem (4) must satisfy three conditions: the feasibility condition, the non-negativity condition, and the complementary slack condition, expressed as follows:

$$
\begin{aligned}
H(s) &\leq 0, \\
\lambda &\geq 0, \\
\lambda \cdot H(s) &= 0.
\end{aligned}
\tag{16}
$$

Given that $H(s) > 0, \forall s \in S$, the primal feasibility condition is violated; there are no feasible solutions that satisfy $H(s) \leq 0$. The complementary slackness condition requires that for each $s \in S$, either $\lambda = 0$ or $H(s) = 0$. Since $H(s) > 0$, we must have $\lambda = 0$ to satisfy this condition. However, setting $\lambda = 0$ does not penalize the constraint violation, and the primal feasibility condition remains unsatisfied. This leads to a contradiction: there is no finite $\lambda \geq 0$ that satisfies all the KKT conditions when $H(s) > 0, \forall s \in S$. The optimization problem is infeasible because the constraints cannot be met. In practical online training, the Lagrange multiplier $\lambda$ is adjusted iteratively to enforce the constraints by increasing $\lambda$ whenever the constraints are violated. Since $H(s) > 0$ always holds, the Lagrange multiplier $\lambda$ will continually increase in an attempt to penalize the constraint violations. As a result, $\lambda$ can become arbitrarily large, theoretically approaching infinity.

□

## D.3 HETEROGENEOUS GRAPH NETWORK

To encode the topological and attribute information of $G_I$, we enhance widely-used graph attention networks (GAT) (Veličković et al., 2018) by integrating heterogeneous link fusion and link attribute encoding in the propagation process. We begin by using the MLPs to generate the representations of virtual and physical nodes, i.e., $H_v^0 = \mathrm{MLP}(X_v^n), H_p^0 = \mathrm{MLP}(X_p^n)$. Then, we leverage $K$ layers of GNNs to assimilate the topological and bandwidth information. At each $\kappa$-th layer, a link feature-aware GAT extracts all node latent representations independently across different link types:

$$
\bar{H}_v^\kappa = \mathrm{GAT}(H_v^{\kappa-1}, L_v, X_v^l), \quad \bar{H}_p = \mathrm{GAT}(H_p^{\kappa-1}, L_p, X_p^l),
\tag{17}
$$

$$
Z_{v,m}^\kappa, Z_{p,m}^\kappa = \mathrm{GAT}\left(\left[Z_v^{\kappa-1}; Z_p^{\kappa-1}\right], L_{v,p,m}, X_{v,p,m}^l\right),
\tag{18}
$$

$$
Z_{v,d}^\kappa, Z_{p,d}^\kappa = \mathrm{GAT}\left(\left[Z_v^{\kappa-1}; Z_p^{\kappa-1}\right], L_{v,p,d}, X_{v,p,d}^l\right).
\tag{19}
$$

Here, ; denotes the combination operator. Particularly, when calculating the attention weight between node $i$ and $j$ using their representation $h_i$ and $h_j$, we also apply the MLP to their link attribute $x_{i,j}^l$, i.e., $h_{i,j} = \mathrm{MLP}(x_{i,j}^l)$, and incorporate it into this process to perceive bandwidth resources:

$$
a_{i,j} = \frac{\exp(\mathrm{MLP}(z_i + z_j + z_{i,j}))}{\sum_{k \in \mathcal{N}(i)} \exp(\mathrm{MLP}(z_i + z_k + z_{i,k}))},
\tag{20}
$$

where $\mathcal{N}(i)$ indicates the neighbor set of node $i$.

Then, to aggregate the diverse information presented through different link perspectives, we apply the sum pooling method to produce the node representations for each GNN layer:

$$
Z_v^\kappa = \bar{Z}_v^\kappa + Z_{v,m}^\kappa + Z_p^\kappa, \quad Z_p^\kappa = \bar{Z}_p^\kappa + Z_{p,m}^\kappa + Z_{p,d}^\kappa.
\tag{21}
$$

Finally, leveraging the final layer representations, $H_v^K$ and $H_p^K$, alongside residual connections to bolster initial feature representation, we obtain the final representations of all nodes $Z = \{Z_v, Z_p\}$:

$$
Z_v = Z_v^K + Z_v^0, \quad Z_p = Z_p^K + Z_p^0.
\tag{22}
$$

### D.4 BARLOW TWINS LOSS FUNCTION

In this work, we utilize the contrastive learning method to bring the representations under augmented views close to enhance the awareness of bandwidth-constrained path connectivity. One of the main directions of contrastive learning is to utilize both positive and negative sample distinctions (Oord et al., 2018; Chen et al., 2020). However, in our application, generating and selecting negative samples that are markedly different in terms of feasibility semantics is significantly challenging. This difficulty can adversely affect the learning quality and the generalizability of the models. Therefore, considering multiple views of the graph with the same feasibility semantics, we focus on eschewing negative samples altogether while preventing feature collapse, an emerging direction of contrastive learning (Grill et al., 2020; Zbontar et al., 2021). Specifically, we adopt the Barlow Twins method (Zbontar et al., 2021) for its simplicity and effectiveness, which circumvents negative sample selection and maintains the original network architecture. Given the node embeddings under two augmented views, $H^a$ and $H^b$, we reduce redundancy between embedding components by aligning their cross-correlation matrix with the identity matrix:

$$L_{CL} = \sum_i \left(1 - \mathbb{C}_{ii}\right)^2 + w \sum_i \sum_{i \neq j} \mathbb{C}_{ij}^2, \tag{23}$$

where the first one is the invariance term and the second one is the redundancy reduction term. $w$ is tradeoff weight. $\mathbb{C}_{ij}$ is the cross-correlation matrix computed between the output node representations of the two identical networks along the batch dimension: $\mathbb{C}_{ij} = \frac{\sum_b Z_{b,i}^A Z_{b,j}^B}{\sqrt{\sum_b \left(Z_{b,i}^A\right)^2} \sqrt{\sum_b \left(Z_{b,j}^B\right)^2}}$. Here, $b$ indexes node samples. $i, j$ index the node representation dimension, respectively. Subsequently, we will equip our final optimization objective with this unsupervised learning loss function.

### D.5 LAGRANGIAN-BASED PPO TRAINING METHOD.

To optimize the policy for VNE solving, we adopt the Lagrangian-based actor-critic framework with PPO (Ray et al., 2019) objective as our training algorithm, which incorporates constraint guarantees within the RL training process (Schulman et al., 2017). In practical implementation, we derive the node representations $Z_v$ and $Z_p$ from the state $s$, with our constraint-aware graph representation method. These representations serve as inputs to a policy network, which generates an action distribution, $\pi(s) = \text{MLP}(Z^p) \in \mathbb{R}^{|N_p|}$, used for the action selection. Additionally, we employ three additional networks: the value critic network $V_r^\pi$, the reachability critic network $V_h^\pi$, and the Lambda network $\Lambda^\pi$. They have similar architectures, $\forall V \in \{V_r^\pi, V_h^\pi, \Lambda^\pi\}, V(s) = \text{MLP}\left(\sum_{z \in Z_p} z\right) \in \mathbb{R}^1$. They estimate cumulative rewards, constraint violations, and the Lagrangian multiplier $\lambda$, respectively. We optimize $V_r^\pi$ and $V_h^\pi$ with mean squared error (MSE) losses, denoted as $L_r$ and $L_h$, comparing predicted values against actual results for cumulative rewards and violations, respectively. The Lambda network (Ma et al., 2021) is updated to optimize the balance between performance and safety, according to:

$$L_{LAM} = \Lambda^\pi(s) \cdot (V_h^\pi(s) - D_h^{\pi'}(s)). \tag{24}$$

The objective function for the policy network within the PPO framework is expressed as:

$$L_{PPO}(\pi) = \mathbb{E}[\min(r(\pi)A_t, \text{CLIP}(r(\pi), 1 - \epsilon, 1 + \epsilon)A_t)], \tag{25}$$

where $A_t$ represents the advantage function at time step $t$, and $r(\pi) = \frac{\pi(s_t, a_t)}{\pi^{old}(s_t, a_t)}$ calculates the ratio of the probabilities for the selected action between the current and previous policies. The CLIP function limits policy updates to enhance stability.

Finally, integrating the unsupervised contrastive learning objective $L_{CL}$ in the path-bandwidth contrast method, our comprehensive loss function in the training process is formulated as follows:

$$L = w_{PPO} \cdot L_{PPO} + w_r \cdot L_r + w_h \cdot L_h + w_{LAM} \cdot L_{LAM} + w_{CL} \cdot L_{CL}, \tag{26}$$

where all $w_{(\cdot)}$ denote the weights of loss objectives.

### D.6 Detailed Analysis of Computational Complexity

CONAL exhibits a computational complexity of $O\left(|N_v| \cdot K \cdot \left(|L_p|d + |N_p + N_v|d^2\right)\right)$, which the complexities other baseline methods based on RL and GNNs are $O\left(|N_v| \cdot K \cdot \left(|L_p|d + |N_p|d^2\right)\right)$. Here, $N_v$ and $L_v$ denote the number of virtual nodes and links, $N_p$ and $L_p$ denote the number of physical nodes and links, $K$ denotes the number of GNN layers, and $d$ denotes the embedding dimension. Concretely, When constructing a solution for one VNE instance, CONAL performs inference $N_v$ times with the GNN policy, similar to most RL and GNN-based methods. The difference in complexity between CONAL and RL/GNN-based baselines mainly arises from the different neural network structures used, such as GAT and GCN. One GAT and one GCN layer have the same complexity, both $O\left(|L|d + |N|d^2\right)$, where $|N|$ and $|L|$ denote the number of nodes and links [a]. In CONAL, we enhance the GAT with the heterogeneous modeling for the interactions of cross-graph status. Each heterogeneous GAT layer consists of three types of GAT layers for VN, PN, and cross-graph interactions (the number of links between virtual and physical nodes is always $N_p$). The complexities for these layers are $O\left(|L_v|d + |N_v|d^2\right)$, $O\left(|L_p|d + |N_p|d^2\right)$, and $O\left(|N_p|d + |N_p + N_v|d^2\right)$, respectively. Each heterogeneous GAT layer consists of three types of GAT layers for VN, PN, and cross graph (the number of links between virtual and physical node always is $N^p$), whose complexity is $O\left(|L_v|d + |N_v|d^2\right)$, $O\left(|L_p|d + |N_p|d^2\right)$, and $O\left(|N_p|d + |N_p + N_v|d^2\right)$. Considering that $|N_v|$ is significantly smaller than $|L_v|$ and typically $L_p > N_p$ in practical network systems, the overall complexity of CONAL is $O\left(|N_v| \cdot K \cdot \left(|L_p|d + |N_p + N_v|d^2\right)\right)$. In comparison, other RL and GNN-based methods separately encode VN and PN with GAT or GCNs, without considering GNN layers for cross-graph interactions, leading to their complexities being $O\left(|N_v| \cdot K \cdot \left(|L_p|d + |N_p|d^2\right)\right)$. Overall, while CONAL slightly increases the complexity compared to existing RL and GNN-based methods due to its heterogeneous modeling approach, it achieves significant performance improvements.

## E Descriptions of Training and Inference Process

In this section, we summarize the training process of CONAL in Algorithm 1 and the inference process of CONAL in Algorithm 2.

### E.1 Training Process of CONAL

The training process of CONAL begins by randomly initializing the neural networks: the policy network $\pi$, value critic network $V_r^\pi$, feasibility critic network $V_h^\pi$, and lambda network $\Lambda^\pi$. The training involves two key stages: experience collection and policy optimization. Through these stages, we iteratively update the neural networks and ultimately learn the policy. This training process is outlined in Algorithm 1.

During the experience collection stage, for each incoming instance $I \in \mathcal{I}$, we utilize the surrogate policy $\pi'$ to preemptively solve this instance $I$. The maximum constraint violation caused by $\pi'$ is regarded as adaptive reachability budgets $D_h^{\pi'}(s)$ for the following states $s$ sampled by main policy $\pi$. At each decision timestep $t$, we build a heterogeneous graph $G_I$ based on the current situation of instance $I$. We also construct two augmented views, $G_I^A$ and $G_I^B$, with the proposed feasibility-consistency augmentations, which will be used in the calculation of contrastive loss. The state $s_t$ is then extracted from $G_I$, $G_I^A$ and $G_I^B$. Based on the state $s_t$, the policy $\pi$ extracts information with the heterogeneous graph network and selects an action $a_t$. Then, the environment executes this action, transits into the next state $s_t$, and returns the reward $r_t$, violations $h_t$ and costs $c_t$. These items collectively form a transition $(s_t, a_t, r_t, s_{t+1}, h_t, c_t, D_h^{\pi'}(s_t))$ and stored in a trajectory memory $\tau$.

In the subsequent policy optimization stage, we iteratively sample batches of transitions from the trajectory memory $\tau$ to update neural networks. Then, we calculate various losses, including the PPO loss $L_{PPO}$, value critic loss $L_r$, reachability critic loss $L_h$, and Lambda network loss $L_{LAM}$. Additionally, we integrate our proposed unsupervised contrastive loss $L_{CL}$ in path-bandwidth contrast during training. Overall, we calculate the weighted sum of these losses as the final loss $L$ and update neural networks. If necessary, we synchronize surrogate policy $\pi'$ with the main policy $\pi$.

### E.2 Inference Process of CONAL

---

**Algorithm 1** Training process of CONAL.

---

1: **Input:** A set of VNE instances $\mathcal{I}$
2: **Output:** A learned policy $\pi$
3: Initialize the policy network $\pi$, value critic network $V_r^\pi$, feasibility critic network $V_h^\pi$, and lambda network $\Lambda^\pi$ with random weights
4: Initialize the surrogate policy network $\pi'$ same as the policy network $\pi$
5: **# Stage 1: Experience Collection**
6: **for** each VNE instance $I \in \mathcal{I}$ **do**
7:   Compute adaptive reachability budget $D_h^{\pi'}(s)$ for the following sampled state $s$ using $\pi'$
8:   Initialize state $s_0$ from the heterogeneous graph $G_I$ and its augmented views $G_I^A$ and $G_I^B$
9:   **for** timestep $t = 0$ to $T$ **do**
10:     Generate the action probability distribute and select action $a_t \sim \pi(\cdot|s_t)$
11:     Execute action $a_t$, observe reward $r_t$ and next state $s_{t+1}$
12:     Compute constraint violation $h_t = H(s_{t+1})$ and cost $c_t = C(s_{t+1})$
13:     Store transition $(s_t, a_t, r_t, s_{t+1}, h_t, c_t, D_h^{\pi'}(s_t))$ in trajectory memory $\tau$
14:   **end for**
15: **end for**
16: **# Stage 2: Policy Optimization**
17: **for** each update step **do**
18:   Sample a batch of transitions from the trajectory memory $\tau$
19:   Calculate the PPO objective $L_{PPO}$ for policy $\pi$ with Eq. 25
20:   Calculate the value critic loss $L_r$ and reachability critic loss $L_h$
21:   Calculate the loss of Lambda network with Eq. 24
22:   Calculate the contrastive loss $L_{CL}$ with the Barlow Twins method, i.e., Eq. 23
23:   Obtain the final loss with Eq. 25 and update $\pi$, $V_r^\pi$, $V_h^\pi$, and $\Lambda^\pi$
24:   **if** update the surrogate policy **then**
25:     Synchronize the parameters of $\pi'$ with $\pi$, i.e., $\pi' \leftarrow \pi$
26:   **end if**
27: **end for**
28: **return:** The learned policy $\pi$

---

**Algorithm 2** Inference process of CONAL.

---

1: **Input:** An arrived VNE instance $I$; The learned policy $\pi$
2: **Output:** The solution status
    Initialize state $s_0$ from the heterogeneous graph $G_I$
3: **for** timestep $t = 0$ to $T$ **do**
4:   Generate the action probability distribute and select action $a_t \sim \pi(\cdot|s_t)$
5:   Execute action $a_t$: Map the to-be-decided virtual node $n_v^t$ onto the selected physical node $a_t$
6:   Transit to next state $s_{t+1}$: Route all prepared incident links $n_v^t$
7:   Compute constraint violation $h_t = H(s_{t+1})$ and cost $c_t = C(s_{t+1})$
8:   **if** any constraints are violated, i.e., $c_t > 0$ **then**
9:     **return** FALSE
10:   **end if**
11: **end for**
12: **return** TRUE

---

During the inference process, we use the learned policy $\pi$ to solve newly arrived VNE instances. Note that the surrogate policy and the path-bandwidth contrast module are not utilized in this process. The inference process is outlined in Algorithm 2.

At each decision timestep $t$ in the inference process, we attempt to place the virtual node $n_v^t$, and route its prepared incident links $\delta'(n_v^t)$, until the solution is successfully completed or any constraints are violated. Concretely, we extract the state features $s_t$ from the heterogeneous graph, and input them into the policy network to produce an action probability distribution $\pi(a_t \mid s_t)$. The action $a_t$, representing the selected physical node, is chosen using a greedy strategy that picks the action with the highest probability. Then, we execute the selected action, i.e., mapping the virtual node $n_v^t$ onto the physical node $a_t$. Once the action is executed, the network system transitions to the next state $s_{t+1}$, where all prepared incident links $\delta'(n_v^t)$ are routed. Subsequently, the system computes the corresponding constraint violations $h_t$ and costs $c_t$ for the current state. If any constraints are violated during the process, the inference is terminated early, and the instance is rejected. Otherwise, the process continues until a complete and feasible solution is found.

# F  EXPERIMENTAL DETAILS

In this section, we provide the details of implementations and hyperparameter settings, the descriptions of compared baselines and CONAL's variations, and the definition of metrics.

## F.1  IMPLEMENTATION DETAILS

**CONAL Implementation.** We implement the GNNs in CONAL with PyG and other neural networks of CONAL with PyTorch. Each neural network has a hidden dimension of 128 and GAT modules are composed of 3 layers. We set the both reward and cost discounted factor $\lambda$ of CMDP to 0.99. We set the augment ratio $\epsilon$ in the path-bandwidth contrast method to 1. We use a batch size of 128 and the Adam optimizer (Kingma & Ba, 2014) with a learning rate of 0.001. We use the sampling strategy and greedy strategy for the action section in the training and testing processes, respectively. For the $k$-shortest path algorithm used in link routing, we set the maximum path length $k$ to 5. The loss weights are set as follows: $w_{PPO} = 1.0$, $w_{LAM} = 0.1$, $w_{CL} = 0.001$, $w_r = 0.5$ and $w_h = 0.5$.

**Simulation for Training and Testing.** For RL-based methods and CONAL, we train policies for each average arrival rate $\eta$, where running seeds are randomly set in every simulation. During testing, we evaluate the performance of all algorithms by repeating the tests with 10 different seeds (i.e., $0, 1111, 2222, \cdots, 9999$) for each average arrival rate $\eta$ to ensure statistical significance.

**Computer Resources.** All experiments were conducted on a Linux server equipped with one NVIDIA A100 Tensor Core GPU, 24 AMD EPYC 7V13 CPUs, and 128GB of memory.

## F.2  BASELINE DESCRIPTIONS

We introduce the compared baselines, which cover the most perspectives of VNE solving strategies:

- **NRM-VNE** (Zhang et al., 2018) is a node ranking-based heuristic method. It first uses a Node Resource Management (NRM) metric to rank both virtual and physical nodes and employs a greedy matching approach for node mapping. Then, for link mapping, this method utilizes the $k$-shortest path algorithm, similar to our approach.

- **GRC-VNE** (Gong et al., 2014) is a node ranking-based heuristic method. It sorts nodes with a Global Resource Control (GRC) strategy based on random walk and maps them accordingly. Then, it conducts the link mapping using $k$-shortest path algorithm.

- **NEA-VNE** (Fan et al., 2023) is a node ranking-based heuristic that employs a Node Essentiality Assessment (NEA) metric to rank nodes and follows a similar mapping as NRM-VNE and GRC-VNE.

- **GA-VNE** (Zhang et al., 2019) is a meta-heuristic method based on genetic algorithms. It models each node mapping solution as a chromosome and iteratively explores the solution space by simulating the process of natural selection and genetic evolution.

- **PSO-VNE** (Jiang & Zhang, 2021) is a meta-heuristic method that employs particle swarm optimization. It explores the VNE solution space by simulating the behavior of particles.
- **MCTS-VNE** (Haeri & Trajković, 2017) is a model-based RL method. It utilizes the Monte Carlo Tree Search (MCTS) algorithm to explore possible solutions with upper confidence bound strategies.
- **PG-CNN** (Zhang et al., 2022) is a model-free RL method. It models the solution construction of each VNE instance as MDP. Then, it develops a policy network with Convolutional Neural Network (CNN) and trains it using the Policy Gradient (PG) algorithm. Specifically, during training, only samples related to feasible solutions are used for optimization.
- **A3C-GCN** (Zhang et al., 2023b) is a model-free RL method. It constructs a policy network with a Graph Convolutional Network (GCN) and a Multi-Layer Perceptron (MLP). The Asynchronous Advantage Actor-Critic (A3C) algorithm is used for training. Particularly, during training, if encountering failure samples, customized negative rewards are returned.
- **DDPG-ATT** (He et al., 2023a) is a model-free RL method that builds an ATTention-based (ATT) policy network and uses the Deep Deterministic Policy Gradient (DDPG) algorithm for training. Similar to A3C-GCN, it introduces the penalty into reward shaping.
- **GAL-VNE** (Geng et al., 2023) is a model-free RL method that formulates VNE into both Global learning across requests And Local prediction within requests (GAL). It first uses supervised learning to develop a GNN policy to solve VNE in one shot. Then, in online perspective, they use the RL to finetune this policy to improve the overall performance.

Regarding the baseline implementation, we use the official code of GAL-VNE and reproduce DDPG-ATT following the original paper. For the other baselines, we derive their implementations from the Virne[1] library. Furthermore, for their hyperparameter settings, we follow the original papers for heuristic baselines. For RL-based baselines, we set the same hidden size of their neural networks as us and other hyperparameters according to their original papers. In the link mapping process, all baselines use the same $k$-shortest path algorithm as ours.

### F.3 VARIATIONS DESCRIPTIONS

To verify the individual performance contributions of each CONAL's components, we design the following variations as additional baselines:

- **CONAL$_{\text{w/o HM}}$** discards the proposed heterogeneous modeling (HM) module. Instead, it independently extracts the features of VN and PN using GAT. The global representation of VN is obtained using the sum pooling method. Then, we produce the final node representation of PN by adding this global representation, which is enhanced with the path-bandwidth contrast module and used to produce the final action probability distribution.
- **CONAL$_{\text{w/o PC}}$** removes the proposed path-bandwidth contrast (PC) module.
- **CONAL$_{\text{w/o HM \& PC}}$** omits both the HM and PC modules, utilizing independent feature extractions of VN and PN and an addition-based fusion method, same as CONAL$_{\text{w/o HM}}$.
- **CONAL$_{\text{w/o REACH}}$** replaces our reachability-guided optimization objective (i.e., Eq. 3) with the traditional cumulative cost optimization objective (i.e., Eq. 2), which restricts expected cumulative costs below zero. It extend the proposed adaptive reachability budget, i.e., Eq. 5, to calculate adaptive cost budget, defined as follows:

$$\forall s \in \tau, D_c^{\pi'}(s) = \sum_{s' \in \tau'} C(s').$$ (27)

- **CONAL$_{\text{w/o ARB}}$** removes the proposed adaptive reachability budget (ARB) module and uses zero as a fixed reachability budget.

### F.4 METRIC DEFINITIONS

We provide detailed definitions of key evaluation metrics that are widely used to evaluate the effectiveness of VNE algorithms comprehensively (Fischer et al., 2013; Yan et al., 2020):

---
[1]https://github.com/GeminiLight/virne

- VN Acceptance Rate (**VN_ACR**) measures the proportion of VN requests that are successfully accepted by the system. It evaluates the ability of network provider to satisfy user service requests, defined as

$$\text{VN\_ACR} = \frac{\sum_{t=0}^{\mathcal{T}} |\tilde{\mathcal{I}}(t)|}{\sum_{0}^{\mathcal{T}} |\mathcal{I}(t)|}, \tag{28}$$

  where $\mathcal{I}(t)$ and $\tilde{\mathcal{I}}(t)$ denote the set of totally arrived and accepted VNE instances at the unit of time slot $t$. The operation $|\mathcal{I}|$ means the number of VN requests in the set $\mathcal{I}$.

- Long-term Revenue (**LT_REV**) evaluates the total revenue generated over a specified period, serving as an indicator of the financial gains derived from the VN requests processed by the system. It reflects the economic impact of decisions on network operations.

$$\text{LT\_REV} = \sum_{t=0}^{\mathcal{T}} \sum_{I \in \tilde{\mathcal{I}}(t)} \text{REV}(E) \times \varpi, \text{ where } E = f_G(I). \tag{29}$$

  Here, $\varpi$ the lifetime of the tackled VN $G_v$, where $I = \{G_v, G_p\}$.

- Long-term Revenue-to-Consumption Ratio (**LT_R2C**) quantifies the economic efficiency of the system by comparing the revenue generated to the resources consumed. It offers insights into the operational cost-effectiveness of the VNE solutions implemented.

$$\text{LT\_R2C} = \frac{\sum_{t=0}^{\mathcal{T}} \sum_{I \in \tilde{\mathcal{I}}(t)} \text{REV}(E) \times \varpi}{\sum_{t=0}^{\mathcal{T}} \sum_{I \in \tilde{\mathcal{I}}(t)} \text{CONS}(E) \times \varpi}, \text{ where } E = f_G(I). \tag{30}$$

- Average Solving Time (**AVG_ST**) measures the computational efficiency of VNE algorithm in online inference. We define it as the average wall-clock time required to solve a single instance during one simulation, and use second (s) as the time unit.

- Constraint Violation (**C_VIO**) assesses the constraint satisfaction ability of the VNE algorithm, which is used to compare the constraint awareness of CONAL and its variations. It is defined as cumulative constraint violations over all VNE instances in one simulation:

$$\text{C\_VIO} = \sum_{t=0}^{\mathcal{T}} \sum_{I \in (\mathcal{I}(t) - \tilde{\mathcal{I}}(t))} \sum_{s \sim \tau} C(s), \tag{31}$$

  where $\tau$ is the trajectory produced by the policy $\pi$ with greedy selection and $C(s)$ is the caused violations in state $s$. $\sum_{s \sim \tau} C(s)$ means the sum of violations in the trajectory $\tau$.

# G    ADDITIONAL EVALUATION

**UPDATE**

In this section, we present additional experiments to evaluate the generalizability, scalability, and practicability of CONAL, as well as investigate the impact of two key hyperparameters.

## G.1    GENERALIZABILITY STUDY

To evaluate the robustness and adaptability of CONAL's trained policy across various network conditions, we conduct a series of experiments to test its generalizability in different dynamic and fluctuating environments, i.e. , under varying request frequencies and changing resource demands.

**Request Frequency Sensitivity Study**. We analyze the sensitivity of CONAL and other VNE algorithms to varying arrival rates of VN requests by adjusting the value of $\eta$. This manipulation simulates different network system scenarios with varying traffic throughputs. Specifically, we conduct experiments with $\eta$ values ranging from 0.08 to 0.20, in increments of 0.02. The results are illustrated in Figure 5. As request frequency $\eta$ increases, we observe all algorithms exhibit a clear downward trend in VN_ACR. This is mainly because the increase in request frequency intensifies resource competition among VN requests at the same moment, resulting in more rejections of VNs.

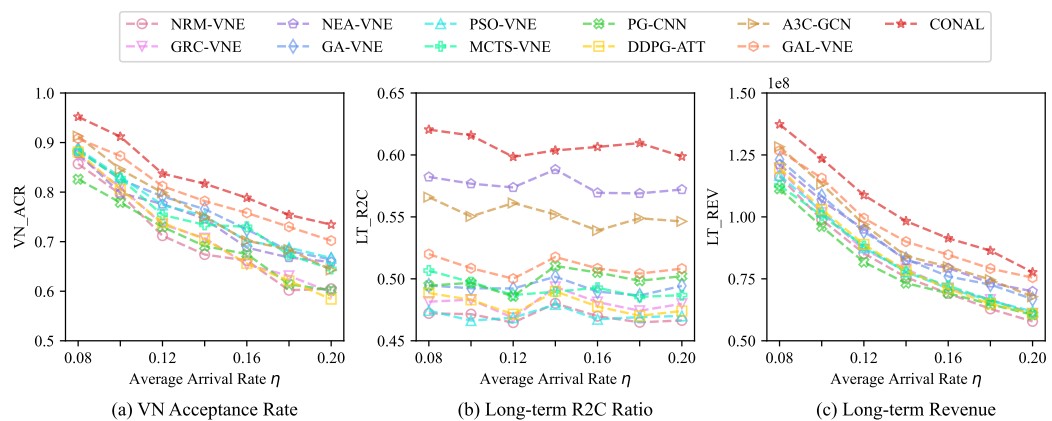

Figure 5: Results in the sensitivity study on varying average arrival rate $\eta$.

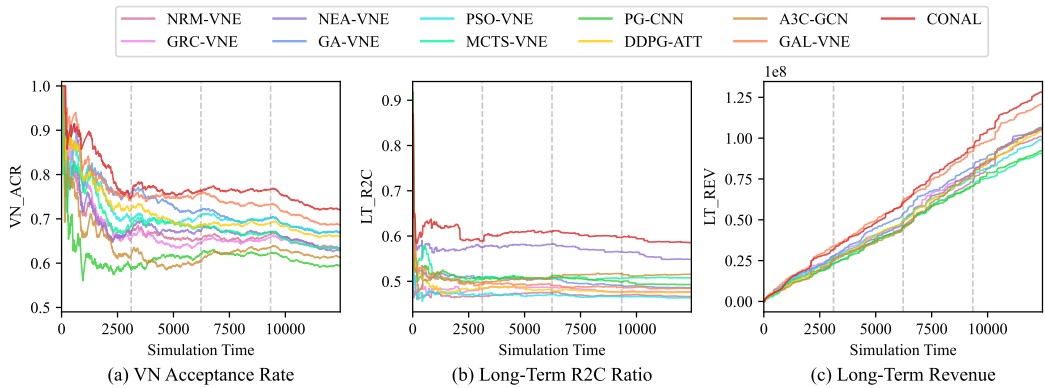

Figure 6: Results in the dynamic request distribution testing. The gray vertical lines roughly partition the request processing stages into four groups with different distributions.

In particular, CONAL consistently outperforms the compared baseline algorithms across all tested values of $\eta$. This indicates that CONAL is more effective and adaptable to changes in request frequency, maintaining superior performance. Overall, this analysis highlights the effectiveness of CONAL in handling complex network environments with fluctuating traffic throughput.

**Dynamic Request Distribution Testing**. In practical scenarios, the node size and resource requirements distributions of VN requests may vary due to different service demands. To simulate such situations, similar to previous work (Geng et al., 2023), we equally divided 1000 VN requests into four sub-groups. In comparison to the default simulation settings, we modified one parameter related to the distribution of VN resource demand or VN node size for each sub-group: For the first group, the node and link resource distributions of VN are changed to [0, 30] and [0, 75], respectively; For the second group, the node and link resource distributions of VN are adjusted to [0, 40] and [0, 100], respectively; For the third group, the VN node size distribution is changed to [2, 15]; For the fourth group, the VN node size distribution is altered to [2, 20]. We evaluate the CONAL and other RL-based methods trained in the default settings. The results are illustrated in Figure 6. We observe that CONAL exhibits superior performance across all metrics, indicating its strong adaptability to dynamic request distributions. Regarding the VN_ACR, all algorithms experience a rapid decrease in the early stages of the simulation due to the quick consumption of resources. Subsequently, their VN_ACR becomes relatively stable. It is worth noting that in the fourth stage, most algorithms show a clear downward trend in VN_ACR. This is because embedding larger VNs with more complex topologies and greater resource demands becomes increasingly challenging. This testing shows that CONAL is highly adaptable across varying VN request distributions, which underscores CONAL's generalization in dynamic network environments.

## G.2 SCALABILITY ANALYSIS

In this section, we assess the scalability of CONAL by exploring its performance in large-scale network environments and its time consumption to adapt to varying network sizes.

**Large-scale Network Validation**. Similar to previous work (Wang et al., 2023), we generate a random Waxman topology (Waxman, 1988) with 500 nodes and nearly 13,000 links, named WX500. This mimics the modern large-scale cloud cluster, which is more challenging due to the increased complexity of the topology. Additionally, we increase the VNE size distribution to a uniform range from 2 to 20 nodes, and set the arrival rate of VN requests $\eta$ to 0.5. All other simulation and hyperparameter settings remain consistent with those outlined in Section 4.1. To adapt the pretrained CONAL model for WX100 to this larger scenario, we fine-tune it on WX500 via transfer learning. The pretraining on WX100 takes approximately 7.274 hours, and the fine-tuning stage on WX500 consumes an additional 4.326 hours. By leveraging transfer learning, we accelerate the training efficiency of CONAL, facilitating the rapid acquisition of a CONAL model suitable for large-scale scenarios. The results are illustrated in Figure 8. We observe that CONAL consistently outperforms the baselines in terms of both VN_ACR and LT_REV. CONAL also demonstrates superior performance in the LT_R2C, which is only marginally lower than NEA-VNE. Regarding the AVG_ST, NEA-VNE, MCTS-VNE, and PSO-VNE exhibit significantly higher time consumption, whereas CONAL maintains a competitive running efficiency similar to PG-CNN, A3C-GCN, and DDPG-ATT. Despite the increased complexity of the WX500 topology, CONAL maintains a balance between performance and computational efficiency. This analysis demonstrates the scalability and efficiency of CONAL in large-scale network scenarios.

**Solving Time Scale Analysis**. We investigate solving time across different physical network sizes to evaluate CONAL's time scalability. Specifically, we increase the size of physical network from 200 to 1,000 nodes with a step of 200 to simulate networks of varying scales. Due to more enough link connectivity of large-scale network system, we set the we set the maximum path length $k$ to 4. Figure 7 shows that as the size increases, NRM-VNE, GRC-VNE and GAL-VNE show the most sightly increased average solving time. In contrast, PSO-VNE and GA-VNE experience a rapid growth in solving time due to their reliance on extensive search. CONAL exhibits the similar trend with A3C-GCN and DDPG-ATT, since they are

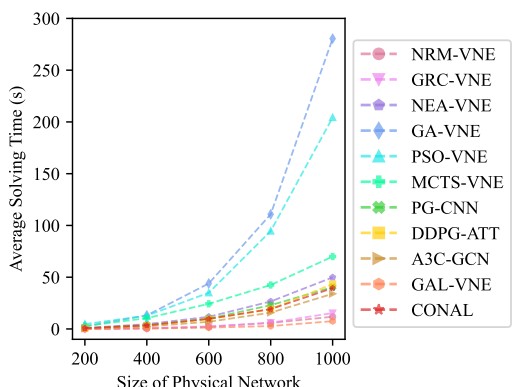

Figure 7: Solving Time Scale with Network Size.

all based on constructive solution paradigm. As the size of physical network increases, the solution time of these algorithms does not explode, still competitive with MCTS and NEA-VNE. Overall, the solving time of CONAL remains efficient even at larger network scales while offering great performance, making it a viable solution for real-time decision-making in large-scale environments.

## G.3 REAL-WORLD NETWORK TOPOLOGY VALIDATION UPDATE

To verify the effectiveness of our proposed algorithm in real-world network topologies, we conducted experiments on realistic network topologies. Similar to previous works (He et al., 2023a; Wang et al., 2023), we employed two widely-used topologies from SDNlib[2]:

- **GEANT** is the academic research network that interconnects Europe's national research and education networks, comprising 40 nodes, 64 edges, and a density of 0.0821.

- **BRAIN** is the largest real-world network topology in SDNlib, consisting of 161 nodes and 166 edges with a density of 0.0129. It is the high-speed data network for scientific and cultural institutions in Berlin.

---

[2]https://sndlib.put.poznan.pl

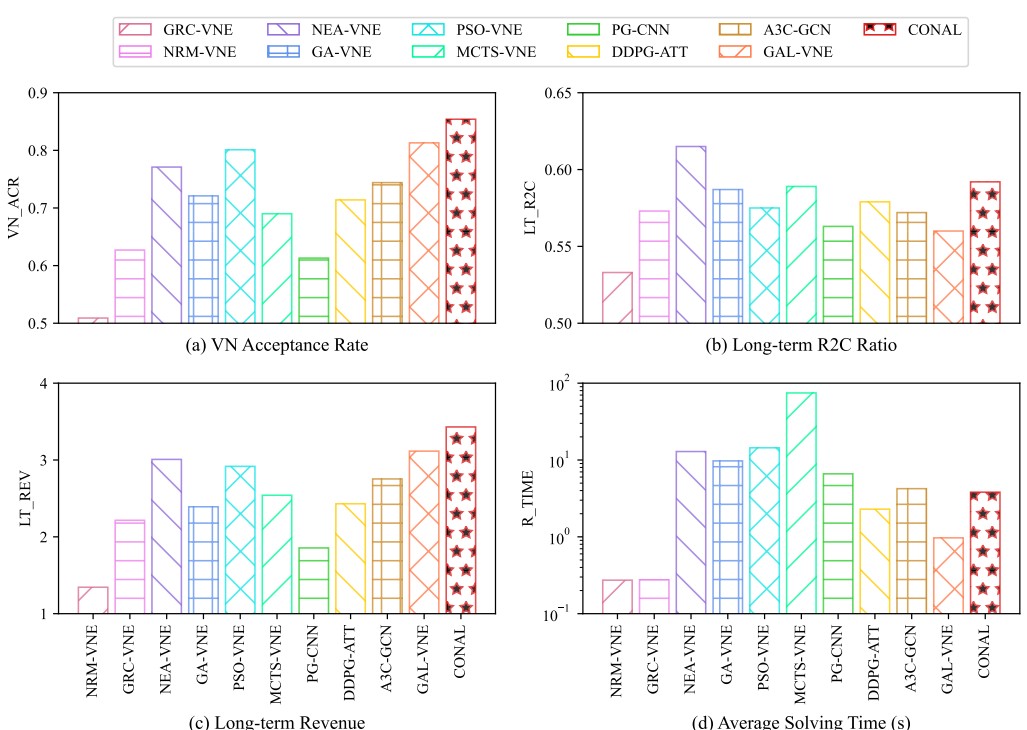

Figure 8: Results in scalability validation.

Table 3: Results in real-world system validation.

| | GEANT | | | BRAIN | | |
|---|---|---|---|---|---|---|
| | VN_ACR ↑ | LT_R2C ↑ | LT_REV ↑ | VN_ACR ↑ | LT_R2C ↑ | LT_REV ↑ |
| NRM-VNE (Fan et al., 2023) | 0.568 | 0.626 | 4.169 | 0.627 | 0.696 | 5.189 |
| GRC-VNE (Gong et al., 2014) | 0.396 | 0.574 | 2.276 | 0.644 | 0.652 | 5.725 |
| NEA-VNE (Fan et al., 2023) | 0.739 | 0.628 | 7.525 | 0.656 | 0.682 | 5.907 |
| GA-VNE (Zhang et al., 2019) | 0.585 | 0.592 | 4.362 | 0.450 | 0.550 | 2.512 |
| PSO-VNE (Jiang & Zhang, 2021) | 0.525 | 0.504 | 3.729 | 0.415 | 0.475 | 2.164 |
| MCTS-VNE (Haeri & Trajković, 2017) | 0.573 | 0.524 | 4.443 | 0.494 | 0.566 | 2.733 |
| PG-CNN (Ma et al., 2023) | 0.639 | 0.576 | 5.712 | 0.538 | 0.651 | 3.804 |
| DDPG-ATT (He et al., 2023a) | 0.625 | 0.533 | 5.449 | 0.572 | 0.657 | 4.326 |
| A3C-GCN (Zhang et al., 2023b) | 0.747 | 0.692 | 8.135 | 0.577 | 0.715 | 4.473 |
| GAL-VNE (Geng et al., 2023) | 0.804 | 0.613 | 9.915 | 0.591 | 0.729 | 4.922 |
| **CONAL** | **0.916** | **0.761** | **11.946** | **0.683** | **0.835** | **6.339** |

* Values in the LT_REV column are scaled by $10^7$.

Due to the limited resource supply in these topologies, we adjust the average arrival rate $\eta$ of VN requests to 0.0005 in GEANT and 0.001 in BRAIN. We keep other network system simulation parameters as same as those discussed in Section 4.1. As shown in Table 3, CONAL outperforms all baselines across both network systems in terms of performance metrics. Notably, in the sparser BRAIN topology, RL-based VNE algorithms (e.g., A3C-GCN and GAL-VNE) performed worse compared to node ranking-based methods (e.g., GRC-GCN and NEA-VNE). The increased challenge of satisfying routing constraints in sparser topologies likely accounts for this performance discrepancy. These RL-based methods with less constraint awareness, result in a low solution feasibility guarantee and tend to exhibit lower performance. Meanwhile, CONAL still achieves the best performance, showing its effectiveness in these real-world network topologies. Furthermore, in the GEANT topology, CONAL achieve an average solving time of just 0.09 seconds, while maintaining efficient performance with an average solving time of 0.2 seconds in the larger-scale BRAIN topology. These results highlight CONAL's capability to deliver rapid solutions, making it highly suitable for low-latency applications in small to medium-sized network environments. **NEW**

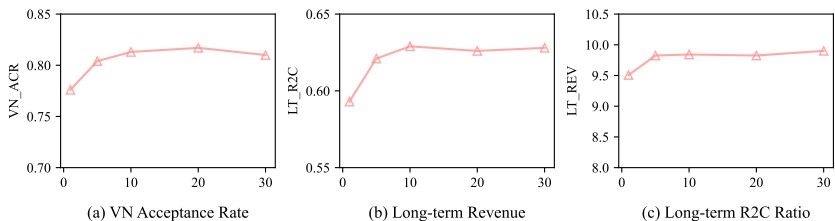

Figure 9: The impact of update interval $\mu$ of surrogate policy proposed in the ARB method.

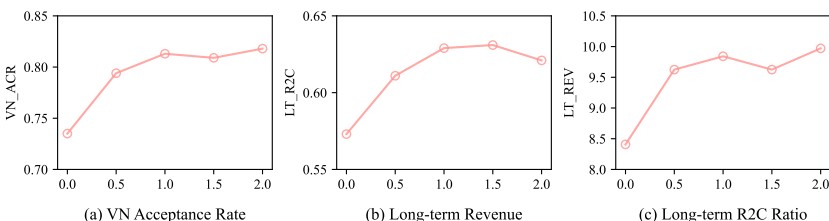

Figure 10: The impact of augment ratio $\epsilon$ used in the path-bandwidth contrast module.

## G.4 HYPERPARAMETER IMPACT STUDY

In this section, we explore the impact of two proposed key hyperparameters on the performance of CONAL, including the update interval $\mu$ of surrogate policy proposed in the ARB method and the augment ratio ($\epsilon$) used in the path-bandwidth contrast module.

**The impact of update interval of surrogate policy.** This parameter $\mu$ controls the update interval of surrogate policy $\pi'$ during training. We vary $\mu$ within the range [1, 5, 10, 20, 30] to explore its impact on key performance metrics, as shown in Figure 9. We observe that increasing $\mu$ initially leads to an improvement in all metrics. This suggests that extremely frequent updates of the surrogate policy make the budget values change rapidly, potentially leading to instability and divergence in the learning process. As $\mu$ increases beyond nearly 10, the improvements across these metrics tend to be stable and show minimal variation. This implies that while a moderate update interval enhances the model's performance, too-slow updates do not offer a significant further improvement on performance metrics and may even increase computational overhead for training.

**The impact of the augment ratio.** This parameter determines the proportion of links to be added based on the number of nodes in the network. We change the augment ratio $\epsilon$ within the range [0, 0.5, 1.0, 1.5, 2.0] and the results are illustrated in Figure 10. As the augment ratio initially increases from 0 to 1.0, we observe improvements across performance metrics. However, when the augment ratio is increased beyond 1.0, these improvements become marginal or even negative. This indicates that excessive enhancement of the graph structure can increase learning difficulty. The increasing disparity between the enhanced and original graph topologies may also negatively impact performance. This study reveals that a reasonable augment ratio $\epsilon$ benefits the model by improving its sensitivity to bandwidth constraints. However, excessively high $\epsilon$ values provide only slight improvements or can even degrade performance. Generally, setting $\epsilon = 1.0$ or a value close to it provides a balanced trade-off between performance enhancement and model robustness.                    NEW

## G.5 CONVERGENCE ANALYSIS OF LAGRANGE MULTIPLIER                    NEW

To evaluate the behavior of the Lagrange multiplier $\lambda$ during training, we conducted experiments under arrival rates $\eta = 0.14$ of VN requests, corresponding to moderate proportions of unsolvable instances. Specifically, we compared the performance of CONAL with and without the ARB mechanism. The Lagrange multiplier $\lambda$ was monitored over 300 training steps on the WX100 topology, with results shown in Figure 11.                    NEW

The results shown in Figure 11 reveal a clear distinction in the convergence behavior of $\lambda$. Without ARB, $\lambda$ diverges rapidly to extreme values. This divergence reflects instability in the optimization process and leads to unreliable policy updates. In contrast, CONAL with ARB effectively stabilizes $\lambda$ throughout training. By dynamically adjusting the feasibility budget, ARB mitigates the impact of unsolvable instances, ensuring stable training even under challenging conditions. These findings demonstrate the crucial role of ARB in preventing numerical instability and enabling robust policy learning in scenarios with high constraint violations.

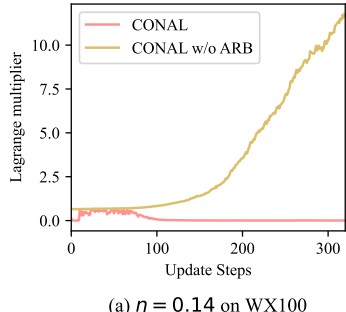

(a) $\eta = 0.14$ on WX100

**NEW**

Figure 11: Convergence curves of Lagrange multiplier $\lambda$ during training.

## H    DETAILED INFORMATION ON USED ASSETS

In this work, we list the used assets along with their version and license as follows:

- **Virne** is a Virtual Network Embedding (VNE) algorithm library, offering many heuristic and machine learning-based methods for VNE. The source code can be accessed at https://github.com/GeminiLight/virne. It is licensed under Apache 2.0. In this work, we derived baseline implementations from Virne and developed our method using this library, specifically version 0.5.0.

- **SNDlib** is a library for telecommunication network design, which offers a collection of realistic network system topologies. This library is open-source and available at https://sndlib.put.poznan.pl, although the specific licensing terms are not clearly stated. In our real-world network validation (see Appendix G.2), we utilize network topologies such as GEANT and BRAIN, which are from SNDlib.

All datasets and codebases are publicly accessible. They focus on networking and communication. They are not directly related to human identities and include no offensive content and bias.

