# OpenReview forum: "Towards Constraint-aware Learning for Resource Allocation in NFV-enabled Networks"
_ICLR.cc/2025/Conference — Submitted to ICLR 2025_

### Official Review · Reviewer_z4vG · 2024-10-28

**Soundness:** 2
**Presentation:** 2
**Contribution:** 2
**Rating:** 5
**Confidence:** 5

**Summary:**

This paper presents a Constraint-aware Network Abstraction Layer (CONAL) tailored for Virtual Network Embedding (VNE) to advance constraint management and improve training robustness, key factors for optimizing network system performance and reliability. By framing VNE as a violation-tolerant Constrained Markov Decision Process (CMDP), the authors aim to enhance solution quality and feasibility, ensuring complete solutions that accurately assess solution quality. The paper introduces a reachability-guided objective, paired with an adaptive feasibility budget method, to guarantee ongoing constraint satisfaction while reducing policy conservativeness and stabilizing policy optimization even with unsolvable instances. To address the complexity of VNE constraints, a constraint-aware graph representation is proposed, featuring a heterogeneous modeling module to capture cross-graph relationships and a path-bandwidth contrast module for heightened sensitivity to bandwidth constraints.

**Strengths:**

The authors propose a violation-tolerant Constrained Markov Decision Process (CMDP) modeling approach, which effectively evaluates solution quality and constraint violation levels, thereby enhancing solution feasibility and resource utilization efficiency.

**Weaknesses:**

- There has been some work examining about RL and NFV [1,2] and various approaches have been devised for solving the constraint violations therein, and it is not clear where this paper excels in relation to them.
  [1] Gu L, Zeng D, Li W, et al. Intelligent VNF orchestration and flow scheduling via model-assisted deep reinforcement learning[J]. IEEE Journal on Selected Areas in Communications, 2019, 38(2): 279-291.
  [2] Zeng Y, Qu Z, Guo S, et al. SafeDRL: Dynamic Microservice Provisioning With Reliability and Latency Guarantees in Edge Environments[J]. IEEE Transactions on Computers, 2023.
- Why applying DRL to cope with VNE is unclear, where existing heuristics do not always face high time overhead and it is not clear for what specific problems face what specific limited performance and why?
- The fact that real-world system validation is not based on real-world system implementations but is still based on simulation should not be blown out of proportion.
- The author claims that reinforcement learning learns effective strategies from unlabeled datasets, however, reinforcement learning actually learns strategies through interaction with the environment.
- Is constraint violation really acceptable for VNE ? Is it reasonable that constraint violations are allowed in the designed solution?

**Questions:**

Compare their work with more existing studies on solving constraint violations in applying RL to NFV （not just the papers mentioned above）, and compare it with them in experiments and related works.

---

> ### Author Response · Authors · 2024-11-24
> **Author Response to W1&Q1 and W2-3**
>
> We sincerely appreciate the time and effort you have dedicated to reviewing our submission and for providing insightful comments and constructive feedback. Below, we address each of your concerns in detail.
>
> > **W1**: There has been some work examining about RL and NFV [1,2] and various approaches have been devised for solving the constraint violations therein, and it is not clear where this paper excels in relation to them.
> >
> > **Q1**: Compare their work with more existing studies on solving constraint violations in applying RL to NFV （not just the papers mentioned above）, and compare it with them in experiments and related works.
>
> Thank you for your suggestion. In the revised manuscript, we have expanded the Related Work section to include comparisons with the mentioned studies and more works. The main related added discussions are as follows
> - Lines 830-837 on Page 16: In particular, Gu et al. (2020) proposed a model-assisted DRL framework that leverages heuristic solutions to guide the training process, reducing reliance on the agent's blind exploration of actions. However, they struggle to handle such complex constraints of VNE thereby compromising performance. Zeng et al. (2024) introduced the SafeDRL algorithm that corrects constraint violations using high-quality feasible solutions through expert intervention. However, this reliance on external corrections ignores the aspects of policy-level constraints awareness, which may limit its adaptability and performance. To address these challenges, we explore learning a constraint-aware VNE policy by innovating existing MDP modeling, representation learning, and policy optimization.
>
> In addition, we will provide the empirical comparison in the subsequent response.
>
> > **W2**: Why use DRL for VNE when heuristics do not always face high time overhead?
>
> Thank you for your question. Heuristic methods have traditionally been favored for VNE due to their simplicity and efficiency; however, they come with significant limitations. Heuristics rely on manually designed rules and expert knowledge, which often fail to generalize across varying network conditions or novel scenarios [a]. Their static nature can lead to suboptimal decision-making, particularly when handling complex trade-offs and constraints inherent in VNE. In contrast, DRL eliminates the need for such manual designs by automatically learning super-heuristics through interaction with the environment [b]. This data-driven approach enables DRL to identify efficient patterns and solutions that are not readily apparent to human designers. Furthermore, DRL dynamically adjusts to changes in network conditions and scales effectively to large and complex networks, making it particularly suitable for real-world VNE scenarios where resource availability and topologies are constantly evolving. Our experimental results highlight that DRL-based methods consistently usually heuristic approaches across critical metrics. Importantly,  the inference phase of DRL exhibits competitive time efficiency with heuristics. As shown in Appendix G.2, the slight increase in time overhead during inference is offset by the substantial gains in solution quality. Overall, while heuristics provide simplicity, their adaptability and performance are limited. DRL addresses these limitations by delivering significant performance improvements while maintaining competitive time efficiency, offering a high-quality solution for VNE.
>
> [a] Yoshua Bengio, et al. Machine Learning for Combinatorial Optimization: a Methodological Tour d'Horizon. EJOR, 2020.
>
> [b] Federico Berto, et al. RL4CO: an Extensive Reinforcement Learning for Combinatorial Optimization Benchmark. NeurIPS GLFrontiers Workshop, 2023.
>
> > **W3**: The fact that real-world system validation is not based on real-world system implementations but is still based on simulation should not be blown out of proportion.
>
> Thank you for your feedback. Following most existing studies [c, d], we have conducted simulation-based evaluations on several real-world network topologies, which is a widely adopted approach in the field. To avoid potential misinterpretation, we have updated the section title in the revised manuscript to "Real-world Network Topology Validation".
>
> [c]  Nan He, et al. Leveraging deep reinforcement learning with attention mechanism for virtual network function placement and routing. TPDS, 2023.
>
> [d] Zhongxia Yan, et al. Automatic Virtual Network Embedding: A Deep Reinforcement Learning Approach with Graph Convolutional Networks. JSAC, 2020.

---

> > ### Comment · Reviewer_z4vG · 2024-11-27
> >
> > Some of our concerns have been addressed. I have slightly increased my score to reflect my improved understanding of your work. However, there are still some ambiguities.1) Why can all constraints be strictly adhered to after training? Is there a theoretical guarantee for this? 2) Why heuristic schemes cannot adapt well to the dynamic changes of the network, while DRL can, is unclear. Why DRL can scale more efficiently to large complex networks is unclear.

---

> ### Author Response · Authors · 2024-11-24
> **Author Response to W4-5**
>
> > **W4**: The claim that reinforcement learning learns effective strategies from unlabeled datasets is misleading. RL learns through interaction with the environment.
>
> We appreciate your feedback and the opportunity to clarify this point. Reinforcement Learning (RL) can learn strategies either through interactions with the environment or from unlabeled datasets, as demonstrated in prior works [e, f]. Both methods could be used to address the VNE problem. Here, we aim to emphasize the distinction between RL and supervised learning, particularly regarding the absence of reliance on labeled data. To avoid any misunderstanding, we have updated the related description as follows:
> "Recently, Reinforcement Learning (RL) has been a potential direction for VNE, which learns effective solving policies without the need of labeled datasets."
> We have updated the text accordingly in the revised manuscript, which is highlighted in blue.
>
> [e] Philip J. Ball, et al. Efficient Online Reinforcement Learning with Offline Data. ICML, 2023.
>
> [f] Tianhe Yu, et al. How To Leverage Unlabeled Data in Offline Reinforcement Learning. ICML, 2022.
>
> > **W5**: Is constraint violation really acceptable for VNE ? Is it reasonable that constraint violations are allowed in the designed solution?
>
> Thank you for your feedback. VNE requires strict adherence to zero constraint violations. In CONAL, we only enable constraint violation tolerance during the training phase to generate complete solutions where constraints may otherwise be unsatisfiable. This approach facilitates a precise evaluation of solution quality and constraint violation degrees, which helps us learn effective strategies for handling highly constrained VNE scenarios. However, as stated in Lines 225–227 (Section 3.1, Page 5) of the manuscript, constraint violations are not permitted during inference. Once the policy has been trained, it operates under strict adherence to all constraints, ensuring feasibility and compliance in deployment scenarios.
>
> Again, we thank the reviewer for your in-depth suggestions for improving our submission. We hope the responses address your concerns. Thank you for your time and consideration.

---

> ### Author Response · Authors · 2024-11-24
> **Author Further Response to Q1**
>
> > **Q1**: Compare their work with more existing studies on solving constraint violations in applying RL to NFV （not just the papers mentioned above）, and compare it with them in experiments and related works.
>
> Thank you for your question. To provide a more comparison analysis, we incorporated the mDDPG  (JSAC, 2019) and the latest SafeDRL method (TC, 2023) proposed in [b] in the experiments. The mDDPG method leverages heuristic solutions to guide the training process and mitigate convergence challenges but does not explicitly address constraint violations in the VNE context. On the other hand, SafeDRL focuses on violation correction by using high-quality feasible solutions, specifically in the microservice provisioning task. We adapted both methods to the VNE problem setting and implemented them using the Virne framework to enable a fair comparison.
> The key performance results are summarized below:
>
> | Method     | VN RAC ↑             | LT R2C ↑            | LT REV (×10⁷) ↑ | AVG ST (×10⁻¹ s) ↓ |
> |------------|----------------------|---------------------|-----------------|--------------------|
> | mDDPG | 0.711 ± 0.017        | 0.493 ± 0.004       | 7.976 ± 0.152   | **3.341 ± 0.075** |
> | SafeDRL    | 0.746 ± 0.013        | 0.512 ± 0.004       | 9.007 ± 0.180   | 3.541 ± 0.144      |
> | **CONAL**      | **0.813 ± 0.042**        | **0.614 ± 0.006**       | **9.842 ± 0.091**   | 4.180 ± 0.104     |
>
> The results demonstrate that CONAL consistently outperforms both SafeDRL and mDDPG across key metrics.
> The superior performance of CONAL can be attributed to its tailored constraint-aware design, which addresses the intricate requirements of VNE scenarios.
> Below, we provide a design comparison of the core design features:
>
> | Feature               | CONAL                                              | SafeDRL                                          | mDDPG |
> |-----------------------|----------------------------------------------------|-------------------------------------------------|-----|
> | **MDP Modeling**      | Constrained MDP with violation tolerance           | Traditional MDP Modeling without explicit consideration of constraint                        | Traditional MDP Modeling without explicit consideration of constraint |
> | **Representation**    | Constraint-aware graph representation              | Multi-Layer Perceptron (MLP)                    | Multi-Layer Perceptron (MLP)  |
> | **Optimization**      | Lagrangian PPO with adaptive reachability budget   | DDPG                     | DDPG                     |
> | **Handling Violations** | Proactive through modeling and optimization strategies | Violation correction via expert intervention   | Not explicitly addressed |
>
> In summary, CONAL’s violation-tolerant CMDP modeling, constraint-aware graph representation, and adaptive optimization techniques collectively enable it to address the unique complexities of VNE problem effectively.
>
>
> [a] Gu L, Zeng D, Li W, et al. Intelligent VNF orchestration and flow scheduling via model-assisted deep reinforcement learning[J]. IEEE Journal on Selected Areas in Communications, 2019, 38(2): 279-291.
>
> [b] Zeng Y, Qu Z, Guo S, et al. SafeDRL: Dynamic Microservice Provisioning With Reliability and Latency Guarantees in Edge Environments[J]. IEEE Transactions on Computers, 2023.

---

> ### Author Response · Authors · 2024-11-27
> **Further Response to Reviewer z4vG on NEW-Q1**
>
> Thank you for your thoughtful feedback and for revisiting our work. We are pleased that some of your concerns have been addressed, and we appreciate the opportunity to clarify the remaining ambiguities.
>
> > **NEW-Q1**: Why can all constraints be strictly adhered to after training? Is there a theoretical guarantee for this?
>
> Thank you for your insightful question. While our training method has theoretical guarantees, it is important to note that the trained policy may not always strictly adhere to all constraints under every case. Below, we clarify both the theoretical guarantees of our approach and the potential factors affecting constraint satisfaction in practice:
>
> **Theoretical Guarantees in Training**: Our method employs a Lagrangian PPO-based method with reachability analysis to train a policy with strict adherence to constraints.  Reachability analysis establishes the largest feasible set—a subset of states where constraints can be persistently satisfied [a]. This is achieved through a feasible value function, which quantifies the worst-case constraint violations over time, guiding the learning process to avoid states that risk violating constraints. By incorporating these insights, our method proactively learns policies that operate within this feasible set.
> This Lagrangian-based PPO training method for reachability analysis has been proven through multi-time scale stochastic approximation theory in work [b].  This ensures convergence of the learned policy to a local optimum, where all safety constraints are satisfied within the feasible set.
>
> **Practical Challenges in Adherence**: Despite the theoretical robustness, strict constraint adherence in practice may be influenced by several factors:
> - *Insolvable Instances*: When no feasible solution exists for one instance (e.g., due to overly restrictive constraints or insufficient resources),  strict adherence is inherently unattainable.
> - *Feature Representation Limitations*:  The expressiveness of the feature representation is critical for accurately perceiving state constraints and improving the quality of the solution. Insufficient representations may lead to suboptimal decisions and constraint violations. This underscores the significance of our constraint-aware graph representation, which alleviates this limitation by enhancing the model's ability to represent complex constraints, such as bandwidth feasibility. However, challenges may still persist in scenarios with exceptionally intricate constraint conditions.
> - *RL Optimization Challenges*: Practical RL training often faces challenges such as local minima or insufficient exploration, which may result in suboptimal policies that do not strictly adhere to all constraints. These are well-known limitations in RL and are not unique to our method [c]. While our adaptive reachability budget method mitigates these issues by providing additional stability during training, it still remains an open problem in DRL.
>
> Overall, our approach is trained with theoretical guarantees for operating within the largest feasible set, and addresses key challenges like unsolvable instances, complex constraints, and stable optimization. While practical factors such as feature representation quality and optimization dynamics may affect strict adherence, our advancements in CMDP modeling, constraint-aware graph representation, and adaptive optimization result in a more efficient solution for VNE.
>
> [a] Somil Bansal, et al. Hamilton-Jacobi Reachability: A Brief Overview and Recent Advances. CDC, 2017.
>
> [b] Dongjie Yu, et al. Reachability Constrained Reinforcement Learning. ICML, 2022.
>
> [c] Shangding Gu, et al. A Review of Safe Reinforcement Learning: Methods, Theories and Applications. TPAMI, 2024.

---

> ### Author Response · Authors · 2024-11-27
> **Further Response to Reviewer z4vG on NEW-Q2**
>
> > **NEW-Q2**: Why heuristic schemes cannot adapt well to the dynamic changes of the network, while DRL can, is unclear. Why DRL can scale more efficiently to large complex networks is unclear.
>
> Thank you for your insightful question. Heuristic schemes are typically based on manually designed rules informed by expert knowledge, which inherently limits their quality and adaptability. These rules are often tailored to specific scenarios and lack the flexibility required to handle dynamic environments. For example, network changes such as fluctuating traffic or resource availability often render static heuristics ineffective, as their rule-based nature cannot dynamically adjust to evolving conditions. This rigidity often leads to suboptimal decisions in environments with high dynamics. It is impractical to curate heuristics for all possible scenarios relying on human expertise, due to the overwhelming diversity and unpredictability of real-world conditions.
>
> In contrast, DRL operates on a data-driven learning paradigm to derive effective policies, reducing dependence on manual rule design. DRL excels at handling such complexity and dynamics, which is why it is widely adopted in highly dynamic scenarios like autonomous vehicles [d], robotics [e] and order-dispatching [f]. By interacting with the environment, DRL agents explore and learn policies across diverse conditions. In our work, as mentioned in experimental settings, we train DRL agents in environments simulating a wide range of scenarios, including fluctuating traffic demands and resource availability, ensuring exposure to varying network states. Furthermore, DRL operates on the state-action-reward paradigm, where agents observe the current state and take actions that maximize long-term rewards. This paradigm further enables DRL-based solutions to generalize effectively to unseen situations and dynamically adapt to changing environments.
>
> Regarding the scalability, VNE, as an NP-hard problem, suffers from combinatorial explosion as network size increases. Heuristics, with their static design, struggle to handle this growth due to the exponential increase in complexity. In contrast, DRL leverages neural networks to approximate policies, enabling efficient handling of high-dimensional solution spaces. This capability is crucial for large-scale networks with numerous nodes, links, and dynamic conditions. Integrating GNNs into DRL further enhances scalability by providing structured representations of complex network topologies. This allows DRL-based methods to effectively encode intricate graph-structured constraints and optimize solutions.
>
> Our experiments demonstrate that CONAL and other advanced DRL-based methods often maintain high performance in both dynamic and large-scale network scenarios. This empirical evidence also underscores the scalability and adaptability of DRL compared to heuristic methods in VNE problem.
>
> [d] Shuo Feng, et al. Dense reinforcement learning for safety validation of autonomous vehicles. Nature, 2023.
>
> [e] Chen Tang, et al. Deep Reinforcement Learning for Robotics: A Survey of Real-World Successes. arXiv, 2024.
>
> [f] Zhaoxing Yang, et al. Rethinking Order Dispatching in Online Ride-Hailing Platforms. KDD, 2024.
>
> We hope this response clarifies the remaining ambiguities. We are committed to addressing any further concerns and are grateful for your thoughtful feedback, which continues to improve our work.

---

> ### Author Response · Authors · 2024-11-30
> **Further Response to Reviewer z4vG on NEW-Q2 (DRL for Networking)**
>
> Dear Reviewer z4vG,
>
> We would like to further elaborate on the transformative potential of DRL in the networking domain, particularly its ability to address complex and dynamic challenges. DRL has increasingly demonstrated its effectiveness across a broad range of networking tasks, showcasing adaptability and scalability that meet the intricate demands of modern networks.
>
> The networking community has widely recognized DRL’s success in diverse scenarios, highlighting its capacity to transform how networking challenges are addressed. Notable applications include network planning [a], adaptive multi-timescale scheduling [b], bandwidth-adaptive compression [c], optimizing control frameworks for RANs [d], resource allocation in NFV [e], and so on. These studies consistently demonstrate that DRL outperforms traditional heuristics by learning optimal strategies in dynamic environments. Its ability to adapt to fluctuating conditions, maintain robust performance, and tackle real-world complexities underscores DRL’s significant value in networking research and practical applications.
>
> Collectively, these advancements illustrate DRL’s broad applicability and substantial impact. Unlike static heuristic methods, DRL employs a data-driven learning paradigm, enabling it to dynamically adapt to diverse conditions, optimize resource utilization, and make decisions in complex, real-time scenarios. By interacting with the environment, DRL develops strategies that generalize effectively across varying network states and scales efficiently to handle large and intricate topologies.
>
> In our work, we propose a DRL-based solution with constraint awareness to tackle the VNE problem, a constrained combinatorial optimization challenge. Our method significantly enhances the model’s capability to handle intricate constraints, advancing the state of the art in VNE research. Furthermore, our contributions extend to the broader field of machine learning-driven networking solutions, providing a robust framework for addressing the pressing challenges of modern networks.
>
> We sincerely thank you for your thoughtful feedback. We hope this additional perspective further clarifies your ambiguities. We look forward to engaging in further discussion and receiving your insights.
>
> Sincerely,
>
> The Authors
>
> References
>
> [a] Hang Zhu, et al. Network planning with deep reinforcement learning. SIGCOMM, 2021.
>
> [b] Yijun Hao, et al. EdgeTimer: Adaptive Multi-Timescale Scheduling in Mobile Edge Computing with Deep Reinforcement Learning. INFOCOM, 2024.
>
> [c] Muhammad Osama Shahid, et al. Cloud-LoRa: Enabling Cloud Radio Access LoRa Networks Using Reinforcement Learning-Based Bandwidth-Adaptive Compression. NSDI, 2024.
>
> [d] Azza H. Ahmed, et al. Deep reinforcement learning-based control framework for radio access networks. MOBICOM, 2022.
>
> [e] Zeng Y, et al. SafeDRL: Dynamic Microservice Provisioning With Reliability and Latency Guarantees in Edge Environments. IEEE Transactions on Computers, 2023.

---

> ### Author Response · Authors · 2024-12-02
> **Looking Forward to Further Discussion**
>
> Dear Reviewer z4vG,
>
> We sincerely appreciate your dedicated time and effort in reviewing our submission and providing thoughtful feedback. We greatly value your insights and hope our responses have adequately addressed your concerns.
>
> As the discussion deadline approaches, we would like to kindly invite you to share any additional feedback or questions you may have. We are more than happy to provide further details or clarifications.
>
> Thank you once again for your thoughtful comments. We look forward to hearing any further thoughts you might have.
>
> Best regards,
>
> The authors

---

### Official Review · Reviewer_iEV6 · 2024-10-31

**Soundness:** 3
**Presentation:** 3
**Contribution:** 3
**Rating:** 6
**Confidence:** 4

**Summary:**

The paper proposes a solution based on a Constrained Markov Decision Process for resource allocation in NFV-enabled networks. The problem of resource allocation in those networks (called Virtual Network Embedding in the related literature) is well-known in the research community and several solutions for it have already been proposed. Anyway, the proposed solution is sufficiently original and shows to achieve good performance results if compared with the primary baselines already existing in the literature.
However, the paper is weak in terms of in-depth technical insights about how to efficiently implement the proposed solution, of insufficient experimental evaluation and validation, and of potential impact in the field (see the following parts of this review form).

**Strengths:**

- The addressed topic is interesting and relevant, even if already well investigated in the related literature
- The proposed problem formulation and the deriving algorithmic solution are technically sound and do not exhibit big technical flaws
- The reported performance results are interesting and show that the proposed solution can outperform several related baselines in the existing literature
- The paper is generally well organized and well written

**Weaknesses:**

- The VNE problem has been investigated several times in the related literature. To be impactful, there is the need that novel solutions in the field do not propose only an algorithmic solution but also the design and implementation of a prototype integrated into real cloud/edge deployment environments. Otherwise, the level of technical originality and relevance could only be limited, given the status of maturity of the research field
- The paper does not include in-depth technical insights about how to exactly achieve an effective and efficient design/implementation of the proposed solution into a real prototype. No lessons learned from the experience of real deployment and evaluation in in-the-field deployment scenarios
- No systems engineering considerations and lessons learned about how to optimally configure and deploy the proposed solution
- The reported performance results are obtained by adopting simulation assumptions that are not realistic for many real deployment environments. I can understand that other papers in the literature have adopted a similar approach, but this is too simplistic. At least the validity of the used assumptions should be better justified and motivated in the paper. In addition, why not using real traces from real deployment environments, in particular for request demands?
- Even if the paper is generally well organized and well written, a few writing inaccuracies are still present in the manuscript and call for some minor revision work in order to improve the paper presentation style. Only to mention one example: "Addtional" in page 24.

**Questions:**

Please see the previous parts of this review form, in particular the weaknesses part above.

**Details Of Ethics Concerns:**

N/A.

---

> ### Author Response · Authors · 2024-11-24
> **Author Response to W1-4**
>
> We sincerely appreciate the time and effort you have dedicated to reviewing our submission and for providing insightful comments and constructive feedback. Below, we address each of your concerns in detail.
>
> > **W1**: The VNE problem has been investigated several times in the related literature. To be impactful, there is the need that novel solutions in the field do not propose only an algorithmic solution but also the design and implementation of a prototype integrated into real cloud/edge deployment environments. Otherwise, the level of technical originality and relevance could only be limited, given the status of maturity of the research field
> >
> > **W2**: The paper does not include in-depth technical insights about how to exactly achieve an effective and efficient design/implementation of the proposed solution into a real prototype. No lessons learned from the experience of real deployment and evaluation in in-the-field deployment scenarios
> >
> > **W3**: No systems engineering considerations and lessons learned about how to optimally configure and deploy the proposed solution.
>
> Thank you for your suggestion. Our work, like many algorithm-focused studies on VNE [a,b,c], emphasizes the development and evaluation of a generalizable and robust algorithmic framework. To ensure comparability with existing research, we adopt widely used benchmarks and experimental setups from the literature. While we acknowledge the importance of prototype design and real-world deployment, our focus is on advancing algorithmic innovations, such as RL and GNN, which are of particular interest to the machine learning community, including venues like ICLR. In  future work, we will attempt to deploy our method in real-world settings to bridge the gap between research and practical applications.
>
> [a] Sheng Wu, et al. AI-Empowered Virtual Network Embedding:A Comprehensive Survey. IEEE Communications Surveys & Tutorials, 2024.
>
> [b] Haoyu Geng, et al. GAL-VNE: Solving the VNE Problem with Global Reinforcement Learning and Local One-Shot Neural Prediction. KDD, 2023.
>
> [c] Tianfu Wang, et al. FlagVNE: A Flexible and Generalizable Reinforcement Learning Framework for Network Resource Allocation. IJCAI, 2024.
>
> > **W4**: The reported performance results are obtained by adopting simulation assumptions that are too simplistic and not realistic for many real deployment environments. I can understand that other papers in the literature have adopted a similar approach, but this is too simplistic. At least the validity of the used assumptions should be better justified and motivated. In addition, why not using real traces from real deployment environments, in particular for request demands?
>
> Thank you for your feedback. We understand your concern regarding the reliance on simulated environments and the lack of real-world traces. Unfortunately, to the best of our knowledge, there are currently no publicly available datasets. We have carefully reviewed the literature [a and all works in submission reference] and found that nearly all publications in top journals and conferences addressing the VNE problem similarly rely on simulation benchmarks to evaluate their algorithms. To address potential concerns, we designed our simulation datasets meticulously to closely resemble real-world networking conditions. For instance, we incorporated widely accepted topologies, realistic virtual network request patterns, and diverse resource constraints. Detailed information about our simulation settings and the rationale behind them is provided in the manuscript to ensure transparency and reproducibility. These benchmarks, while simulated, have been crafted to provide a approximation of real-world scenarios and are validated by their widespread adoption in the community. We will also continue to pay attention to whether available real datasets are released in the future and verify our proposed method based on it.
>
> [a] Sheng Wu, et al. AI-Empowered Virtual Network Embedding:A Comprehensive Survey. IEEE Communications Surveys & Tutorials, 2024.

---

> > ### Comment · Reviewer_iEV6 · 2024-11-25
> >
> > Thanks for the kind and detailed rebuttal.
> > I can understand your point about the fact that the majority of published articles have only an algorithmic approach (not aiming at real system integration), but anyway this is now insufficient IMHO, given the relative maturity level reached by the research activities in this field.
> > About the usage of simulated traffic traces, again I can understand the point that most papers adopt this approach. However, some datacenter traces start to be available and could be considered. For example, only via a rapid Google search you can find:
> > https://www.sciencedirect.com/topics/computer-science/pcap-file
> >
> > Anyway, I will consider these elements and slightly revise my review accordingly.

---

> > > ### Author Response · Authors · 2024-11-25
> > > **Response to Reviewer iEV6**
> > >
> > > Dear Reviewer iEV6,
> > >
> > > Thank you for your suggestion and for sharing the [link](https://www.sciencedirect.com/topics/computer-science/pcap-file). We have reviewed the resource link but were unable to find a directly applicable dataset. Additionally, after an extensive search for datasets, we still find that relevant datasets are limited.
> > >
> > > We attribute this to the inherent complexity and emerging characteristics of NFV systems. The datasets required for this domain, particularly for VNE, are not only time-series data related to virtual machines in datacenters but also need to include user networking services represented in a graph format.
> > >
> > > We remain committed to actively seeking real-world datasets that meet these specific requirements for further validation.
> > >
> > > We deeply value your suggestion and sincerely thank you for your prompt response.
> > >
> > > Best regards,
> > >
> > > The authors

---

> ### Author Response · Authors · 2024-11-24
> **Author Response to W5-6**
>
> > **W5**: Insufficient performance results about the latency introduced by the proposed solution. It is true that for large-scale deployment environments the response time of the proposed solution is better than other baselines, but the Authors do not focus on which is exactly the latency in low-medium scale scenarios: for several application domains, a latency of around 2s could be considered excessive and it is not clear, given the scale of the y axis in Figure 8, which are exactly the number of ms that are more common for classical deployment environments. Which limitations stem from that? For which application domains is the proposed solution not feasible?
>
> Thank you for your feedback. VNE is a well-known NP-hard combinatorial optimization problem. As network size and complexity increase, solving time inherently grows due to the problem's computational nature.  In Figure 8, we focus on evaluating the scalability of CONAL in large-scale scenarios, as this represents the most challenging and impactful use cases for real-world deployment. The results demonstrate that CONAL performs well in these scenarios, with better scalability and response times compared to baselines, highlighting its effectiveness in handling large-scale network environments.
> For smaller topologies, the solving time is significantly reduced. As detailed in Table 1, CONAL achieves solving times under 0.5 seconds for WX100 (with 100 nodes and 500 links). Furthermore, for real-world topologies like GEANT and BREAN, which have smaller scales, the solving times are even lower—approximately 0.09 seconds for GEANT (with 40 nodes and 64 links) and 0.2 seconds for BREAN (with 161 nodes and 166 links). These results underline that CONAL's time consumption is well within acceptable ranges for classical deployment environments in smaller-scale scenarios. In the revised manuscript, we have included the solving times for smaller topologies to highlight CONAL’s practical feasibility for low-to-medium scale scenarios in Appendix G.3.
>
> > **W6**: Even if the paper is generally well organized and well written, a few writing inaccuracies are still present in the manuscript and call for some minor revision work in order to improve the paper presentation style. Only to mention one example: "Addtional" in page 24.
>
> We appreciate your attention to detail. We have corrected the specific example you pointed out and thoroughly proofread the manuscript to correct all typographical errors and improve the overall presentation.
>
> Again, we thank the reviewer for your in-depth suggestions for improving our submission. We hope the responses address your concerns. Thank you for your time and consideration.

---

> > ### Comment · Reviewer_iEV6 · 2024-11-25
> > **I am OK with the revised version, thanks for the additional material**
> >
> > Thanks for the kind and detailed rebuttal.
> > I have appreciated the additional material included, in particular the additional quantitative performance measurements.
> > I will consider these elements and slightly revise my review accordingly.

---

### Official Review · Reviewer_9yrG · 2024-11-03

**Soundness:** 3
**Presentation:** 4
**Contribution:** 3
**Rating:** 6
**Confidence:** 3

**Summary:**

The paper tackles the VNE problem within NFV networks. Recognizing the limitations of existing solutions in handling intricate constraints and unsolvable instances, the authors propose a framework called Constraint-Aware Learning, formulates the VNE problem as a violation-tolerant constrained Markov Decision Process and introduces a reachability-guided optimization with adaptive reachability budgets. Additionally, the framework incorporates a constraint-aware graph representation method to capture cross-graph interactions and bandwidth-constrained path connectivity.

**Strengths:**

The paper is well-written and tackles a significant problem, offering practical implications for real-world network systems and potential applicability to other optimization challenges. The performance is benchmarked against several state-of-the-art baselines. Experiments are conducted across a wide range of network scenarios

**Weaknesses:**

The framework seems assumes a static PN setting. This assumption may not hold in highly dynamic network environments, such as mobile edge computing.

The focus is mainly on computing and bandwidth constraints. Other important factors, such as latency, reliability, and energy efficiency, are not addressed.

**Questions:**

1. How does the proposed method perform in dynamic network environments where the physical network topology and resource availabilities could change over time?

2. Can you provide a more detailed analysis of the computational complexity of the proposed method, especially in comparison to baseline methods?

3. Could you elaborate on the rationale behind using contrastive learning in the constraint-aware graph representation module?

---

> ### Author Response · Authors · 2024-11-24
> **Author Response to W1&Q1 and W2**
>
> We sincerely appreciate the time and effort you have dedicated to reviewing our submission and for providing insightful comments and constructive feedback. Below, we address each of your concerns in detail.
>
> > **W1**: The framework seems to assume a static PN setting, which may not hold in dynamic environments like mobile edge computing.
> >
> > **Q1**: How does the proposed method perform in dynamic network environments where PN topology and resource availabilities change over time?
>
> We appreciate your feedback. In this work, we mainly focus on the resource allocation problem in NFV networks, i.e., VNE problem. In dynamic environments like mobile edge computing, the PN topology and resource availabilities can fluctuate over time, introducing additional complexities. This requires additional mechanisms for handling network service migrations, scheduling and backup, which go beyond the scope of the resource allocation task addressed by CONAL.
>
> That said, CONAL's inherent flexibility makes it adaptable to changes in network conditions, regarding both PN and VN. Specifically, CONAL's constraint-aware graph representation can accommodate changes in network topologies and resource availabilities, stemming from the GNN's adaptability and generalizability. This method allows it to handle resource fluctuations effectively at each given snapshot in dynamic network scenarios.
>
> To evaluate CONAL's performance under dynamic conditions without explicitly incorporating additional algorithm designs for migration, scheduling or backup, we conducted experiments using a Dynamic Request Distribution Testing setup (referenced in Appendix G.2). These experiments simulate scenarios where VN topology and resource availability vary dynamically. The results demonstrate that CONAL effectively adapts to resource fluctuations from the VN perspective, highlighting its potential to generalize to dynamic environments, including similar dynamic conditions in PN.
>
> > **W2**: The focus is mainly on computing and bandwidth constraints. Other important factors, such as latency, reliability, and energy efficiency, are not addressed.
>
> Thank you for your feedback. To emphasize the generalizability of proposed method and clarity of our contributions, this work focuses on developing a general framework for managing complex constraints in VNE. Among the various constraints in NFV-enabled networks, bandwidth and computing resources are often the most critical and general and, therefore, form the primary focus of this study. Other factors, such as latency, reliability, and energy efficiency, represent specific variations of the VNE problem. Our framework is flexible and can be extended to incorporate these constraints by adapting the CMDP formulation and the constraint-aware graph representation. We have incorporated these factors as future research directions.

---

> ### Author Response · Authors · 2024-11-24
> **Author Response to Q2**
>
> > **Q2**: Provide a detailed analysis of the computational complexity of the proposed method, especially compared to baseline methods.
>
> Thanks for your feedback. Regarding the analysis and comparison of computational complexity, in our submission, we mentioned these aspects in Section 3.4 and provided a detailed analysis in Appendix 4.6. Below is the relevant content in our submission:
>
> "CONAL exhibits a computational complexity of $O\left(|N_v| \cdot K \cdot \left(|L_p|d + |N_p+N_v| d^2\right)\right)$, which the complexities other baseline methods based on RL and GNNs are $O\left(|N_v| \cdot K \cdot \left(|L_p|d + |N_p| d^2\right)\right)$. Here, $N_v$ and $L_v$ denote the number of virtual nodes and links, $N_p$ and $L_p$ denote the number of physical nodes and links, $K$ denotes the number of GNN layers, and $d$ denotes the embedding dimension. Concretely, When constructing a solution for one VNE instance, CONAL performs inference $N_v$ times with the GNN policy, similar to most RL and GNN-based methods. The difference in complexity between CONAL and RL/GNN-based baselines mainly arises from the different neural network structures used, such as GAT and GCN. One GAT and one GCN layer have the same complexity, both $O\left(|L|d + |N| d^2\right)$, where $|N|$ and $|L|$ denote the number of nodes and links [a]. In CONAL, we enhance the GAT with the heterogeneous modeling for the interactions of cross-graph status. Each heterogeneous GAT layer consists of three types of GAT layers for VN, PN, and cross-graph interactions (the number of links between virtual and physical nodes is always $N_p$). The complexities for these layers are $O\left(|L_v|d + |N_v| d^2\right)$, $O\left(|L_p|d + |N_p| d^2\right)$, and $O\left(|N_p|d + |N_p+N_v| d^2\right)$, respectively. Each heterogeneous GAT layer consists of three types of GAT layers for VN, PN, and cross graph (the number of links between virtual and physical node always is $N^p$), whose complexity is $O\left(|L_v|d + |N_v| d^2\right)$, $O\left(|L_p|d + |N_p| d^2\right)$, and $O\left(|N_p|d + |N_p+N_v| d^2\right)$. Considering that $|N_v|$ is significantly smaller than $|L_v|$ and typically $L_p > N_p$ in practical network systems, the overall complexity of CONAL is $O\left(|N_v| \cdot K \cdot \left(|L_p|d + |N_p+N_v| d^2\right)\right)$. In comparison, other RL and GNN-based methods separately encode VN and PN with GAT or GCNs, without considering GNN layers for cross-graph interactions, leading to their complexities being $O\left(|N_v| \cdot K \cdot \left(|L_p|d + |N_p| d^2\right)\right)$. Overall, while CONAL slightly increases the complexity compared to existing RL and GNN-based methods due to its heterogeneous modeling approach, it achieves significant performance improvements."

---

> ### Author Response · Authors · 2024-11-24
> **Author Response to Q3**
>
> > **Q3**: Elaborate on the rationale behind using contrastive learning in the constraint-aware graph representation module.
>
> Thank you for your feedback. We leverage contrastive learning (CL) in the constraint-aware graph representation module to enhance the model’s awareness of complex constraints, particularly its sensitivity to bandwidth constraints, which are critical yet intricate in VNE scenarios. Below, we elaborate on the rationale:
>
> - Complexity of bandwidth constraints: Bandwidth constraints play a pivotal role in determining solution feasibility, particularly in the context of path routing complexity. At each decision timestep, we need to carefully select a physical node nt p for placing the current virtual node nt v . This selection is dominated by ensuring that feasible connective paths exist to all other physical nodes hosting the virtual node’s neighbors. Here, the feasibility of the path is dominated by the bandwidth availability of physical links to support the bandwidth requirements of all prepared incident links δ′(nt v ). Accurately representing these constraints is vital for generating high-quality embeddings that ensure feasible embeddings for VNE instances.
> - Limitation of Existing GNNs: Conventional GNNs build up on the propagation mechanism that aggregates information along graph links to capture topology information. However, not all physical links contribute positively to this awareness; some with insufficient bandwidths may even introduce noise into node representations. Therefore, these GNNs lack an explicit mechanism to integrate bandwidth constraints, which are essential for perceiving path feasibility. This limitation highlights the need for a more sophisticated approach that incorporates bandwidth awareness into the representation learning process.
> - Motivation for using CL. We utilize CL to address these challenges by explicitly incorporating bandwidth constraint awareness into the graph representation learning process. In the context of bandwidth-constrained VNE, we devise several augmentation methods (e.g., virtual/physical link additions/deletions) to create diverse yet semantically equivalent views of the network graph, introducing variations in connectivity while preserving feasibility. Then, we use the contrastive loss, specifically, Barlow Twins loss, which emphasizes the alignment of representations for bandwidth-feasible paths while penalizing infeasible ones. Theoretically, Barlow Twins minimizes redundancy between embeddings of augmented views by aligning their cross-correlation matrix with the identity matrix [a]. For bandwidth awareness, this mechanism suppresses the influence of irrelevant features (e.g., links with surplus bandwidth) while amplifying features critical to determining bandwidth feasibility in the GNN propagation process [b]. This facilitates embeddings to reflect the feasibility of physical link connectivity, which is critical for effective VNE policies.
> Overall, by considering contrastive learning, the constraint-aware graph representation module achieves enhanced bandwidth sensitivity, and improves a deeper understanding of path feasibility, all of which are critical for addressing the complex constraints of VNE scenarios.
>
> [a] Jure Zbontar, et al. Barlow Twins: Self-Supervised Learning via Redundancy Reduction. ICML, 2021.
>
> [b] Yihao Xue, et al. Investigating Why Contrastive Learning Benefits Robustness against Label Noise. ICML, 2022.
>
> Again, we thank the reviewer for your in-depth suggestions for improving our submission. We hope the responses address your concerns. Thank you for your time and consideration.

---

> > ### Comment · Reviewer_9yrG · 2024-11-26
> >
> > Thank you for your thoughtful and detailed rebuttal to my comments and questions. Most of my concerns have been addressed, and several points of confusion have been clarified. I have slightly increased my score to reflect my improved understanding of your work after reviewing your response and the additional materials provided.
> >
> > I appreciate that your work prioritises real system integration rather than focusing solely on proposing another algorithmic approach, which is a highly commendable motivation. As noted in my previous review, real system implementation prefers testing across diverse scenarios and performance considerations. However, I remain partially unconvinced that the perspectives chosen in this work fully align with the key priorities for real-world NFV networks, particularly given concerns about increased computational complexity.
> >
> > The ML/AI contributions in this work also seem somewhat narrow in scope. A better articulation of how these contributions advance the field and address specific practical challenges would further strengthen the paper and demonstrate its broader impact beyond incremental improvements. While I appreciate the overall effort, my assessment places this work closer to a 6 than an 8, as the system does not allow a score of 7.

---

> ### Author Response · Authors · 2024-11-27
> **Further Response to Reviewer 9yrG**
>
> Thank you for your constructive and thoughtful feedback, as well as for taking the time to reevaluate our work. We greatly appreciate your insightful comments and would like to take this opportunity to clarify and further articulate our contributions to the ML/AI field.
>
> Our work focuses on advancing machine learning for combinatorial optimization (ML4CO), a research area that has garnered significant attention in the ML community, as discussed in Related Work and highlighted in work `[a,b]`. While most existing studies in ML4CO primarily address classical problems such as Traveling Salesperson Problem (TSP) [c,d], Vehicle Routing Problem (VRP) `[e,f]`, Job Shop Scheduling Problem (JSSP) `[g,h]`, and VNE `[i,j]`, they often overlook the complex constraints. However, many real-world applications are modeled as combinatorial optimization (CO) problems with complex constraints that are critical for real-world applicability. Effectively learning and managing these constraints in ML4CO remains a significant and challenging direction that has yet to be actively explored.
>
> In this paper, we aim to push the boundaries of ML4CO by addressing a highly challenging constrained CO problem, i.e., VNE. VNE is characterized by hard and intricate constraints such as cross-graph resource allocation and bandwidth-constrained path routing. Distinct from most prior works, our focus lies in addressing the significant complexity introduced by these constraints and their practical implications. Below, we review our key contributions:
>
> - *Revealing the Impact of Unsolvable Instances*: Through experimental observation and theoretical proof, we highlight the negative impact of unsolvable instances, which are inevitable in practical environments. Our analysis shows that such instances hinder the training process and policy performance, an issue largely overlooked in existing ML4CO research.
> - *Innovative RL Optimization*: To stabilize training in the presence of unsolvable solutions, we propose a novel adaptive reachability budget. This innovation prevents divergence, ensures robust convergence in constrained scenarios, and is easily generalizable to other constrained CO problems.
> - *Rethinking CMDP Modeling*: While existing works often model CO problems directly as MDPs or CMDPs, they simply stop when strict constraints are violated. we address this gap by introducing violation-tolerant CMDP modeling. This enables the complete exploration and precise evaluation of solution space, thus improving performance.
> - *Well-designed Graph Representation*: Capturing complex constraints, particularly bandwidth-constrained path routing, is an unexplored and challenging area in GNN and ML4CO research. To address this, we design novel graph augmentation methods and leverage contrastive learning (CL) to improve bandwidth awareness and constraint representation.
>
> Our work addresses the unique challenges posed by highly constrained CO problems, a less explored area in ML4CO, specifically in VNE. By revealing critical insights, and rethinking CMDP modeling, RL optimization, and GNN representation, we provide a pathway for applying ML to solve more complex and constrained CO problems across diverse domains. We believe our approach extends the frontiers of ML for CO by introducing new paradigms in modeling, optimization, and representation, which are also broadly applicable to other constrained CO Problems beyond VNE.
>
> Thank you again for your feedback, which has been invaluable in reviewing our contributions and articulating their broader impact. We hope this response clarifies the significance of our work and its potential to advance ML4CO and VNE research.
>
> Reference
> > [a] Yoshua Bengio, et al. Machine Learning for Combinatorial Optimization: a Methodological Tour d'Horizon. EJOR, 2020.
> >
> > [b] [Awesome Machine Learning for Combinatorial Optimization Resources](https://github.com/Thinklab-SJTU/awesome-ml4co)
> >
> > [c] Yifan Xia, et al. Position: Rethinking Post-Hoc Search-Based Neural Approaches for Solving Large-Scale Traveling Salesman Problems. ICML, 2024
> >
> > [d] Yimeng Min, et al. Unsupervised Learning for Solving the Travelling Salesman Problem. NeurIPS, 2023.
> >
> > [e] Qingchun Hou, et al. Generalize Learned Heuristics to Solve Large-scale Vehicle Routing Problems in Real-time. ICLR, 2023.
> >
> > [f] Jianan Zhou, et al. Towards Omni-generalizable Neural Methods for Vehicle Routing Problems ICML, 2023.
> >
> > [g] David W Zhang, et al. Robust Scheduling with GFlowNets. ICLR, 2023.
> >
> > [h] Wonseok Jeon, et al. Neural DAG Scheduling via One-Shot Priority Sampling. ICLR, 2023.
> >
> > [i] Tianfu Wang, et al. FlagVNE: A Flexible and Generalizable Reinforcement Learning Framework for Network Resource Allocation. IJCAI, 2024.
> >
> > [j] Haoyu Geng, et al. GAL-VNE: Solving the VNE Problem with Global Reinforcement Learning and Local One-Shot Neural Prediction. KDD, 2023.

---

### Official Review · Reviewer_S9pS · 2024-11-03

**Soundness:** 2
**Presentation:** 2
**Contribution:** 2
**Rating:** 3
**Confidence:** 4

**Summary:**

The paper proposes a new framework called constraint-Aware Learning (CONAL) to address the Virtual Network Embedding (VNE) problem in network virtualization. Specifically, the paper models the VNE problem as a violation-tolerant CMDP and introduces an adaptive reachability budget (ARB) to handle unsolvable instances.

**Strengths:**

The paper models the VNE problem as a violation-tolerant CMDP and introduces an adaptive reachability budget (ARB) to handle unsolvable instances.

**Weaknesses:**

1. The paper models the VNE problem as a violation-tolerant CMDP and introduces an adaptive reachability budget (ARB) to handle unsolvable instances. However, when dealing with unsolvable instances, no policy can satisfy the constraints, the Lagrange multiplier λ may tend to infinity, leading to numerical instability during training. Instability may affect the policy's performance on solvable instances. Provide empirical evidence of the behavior of the Lagrange multiplier λ during training. Specifically, plot the variation of λ over training iterations or time to illustrate how it evolves, especially in the presence of unsolvable instances.
2. The augmentation methods used in the path-bandwidth contrast module (physical link addition ϕA  and virtual link addition ϕB) lack sufficient theoretical and empirical justification. The choice of augmentation ratio ϵ significantly affects model performance, but the paper does not provide detailed analysis or guidelines for selecting these parameters. Provide theoretical explanations for how the augmentation methods contribute to improved bandwidth awareness.
3. The integration of virtual and physical networks into a heterogeneous graph with numerous cross-graph links can lead to information redundancy and noise. Noise from irrelevant links can hinder the model's ability to learn meaningful representations.
4. The experiments are mainly conducted on simulated environments and limited network topologies (e.g., GEANT and BRAIN). This may not adequately demonstrate the model's performance.
5. Given the rapid development of NFV, some relevant literatures are missing to be discussed, e.g., NFVdeep: Adaptive Online Service Function Chain Deployment with Deep Reinforcement Learning, iwqos’19; Adaptive VNF Scaling and Flow Routing with Proactive Demand Prediction, infocom’18; FlexNFV: Flexible Network Service Chaining with Dynamic Scaling, network’19; Joint Optimization of Chain Placement and Request Scheduling for Network Function Virtualization, icdcs’17, etc.

**Questions:**

Overall，when infeasible instances exist in the Virtual Network Embedding (VNE) problem (i.e., there are no embedding solutions that satisfy all constraints), the optimization method employed in the paper causes the Lagrange multipliers (λ) to grow unbounded during training. The unbounded growth of λ leads to overflows or underflows in numerical calculation. The paper does not provide a robust method for detecting infeasible instances, nor does it implement any controls or mitigations for the growth of λ. This oversight means that, when faced with infeasible instances, the model may fail to operate correctly.

---

> ### Author Response · Authors · 2024-11-24
> **Author Response to W1&Q1 and W2**
>
> We sincerely appreciate the time and effort you have dedicated to reviewing our submission and for providing insightful comments and constructive feedback. Below, we address each of your concerns in detail.
>
> > **W1 & Q1**: Provide empirical evidence of the behavior of the Lagrange multiplier λ during training. Specifically, plot the variation of λ over training iterations or time to illustrate how it evolves, especially in the presence of unsolvable instances.
>
> Thank you for your suggestion. We have included an additional analysis of the behavior of λ during training in Appendix G.5. Specifically, we conducted experiments under arrival rates $\eta = 0.14$ of VN requests, corresponding to moderate proportions of unsolvable instances. We compared the performance of CONAL with and without the ARB mechanism. The $\lambda$ was monitored over 300 training steps on the WX100 topology, with results shown in Figure 11. As training progresses, the λ values in CONAL without ARB tend to diverge towards extreme values. In contrast, the results demonstrate that our ARB effectively stabilizes λ, preventing divergence and ensuring robust training while avoiding numerical instability. Please see Appendix G.5 for more details.
>
> > **W2**: The augmentation methods used in the path-bandwidth contrast module (physical link addition ϕA and virtual link addition ϕB) lack sufficient theoretical and empirical justification. The choice of augmentation ratio ϵ significantly affects model performance, but the paper does not provide detailed analysis or guidelines for selecting these parameters. Provide theoretical explanations for how the augmentation methods contribute to improved bandwidth awareness.
>
> Thank you for your feedback. In the path-contrast module, we devise several augmentation methods to create diverse yet semantically equivalent views of the network graph, introducing variations in connectivity while preserving feasibility. Then, we use the contrastive loss, specifically, Barlow Twins loss, which emphasizes the alignment of representations for bandwidth-feasible paths while penalizing infeasible ones. Theoretically, Barlow Twins minimizes redundancy between embeddings of augmented views by aligning their cross-correlation matrix with the identity matrix [a]. For bandwidth constraint awareness, this mechanism suppresses the influence of irrelevant features (e.g., links with surplus bandwidth) while amplifying features critical to determining bandwidth feasibility in the GNN propagation process. This principle has been validated in related studies [b], demonstrating its efficacy in various domains.
>
> This augmentation-contrastive learning framework has been successfully applied in other fields. Typically, existing works design augmentations heuristically or apply stochastic perturbations to generate views while maintaining semantic equivalence, including computer vision [c], knowledge graphs [d], recommendation systems [e], etc. In our case, the augmentation methods (ϕA and ϕB) are carefully designed to create variations of heterogeneous graph while maintaining original feasibility. These augmentations enhance the model's ability to distinguish bandwidth-feasible paths by exposing it to a range of scenarios during training. This approach enables the GNN to learn robust and generalizable representations that prioritize bandwidth-critical features.
>
> Regarding the augmentation ratio ϵ, we provided a detailed analysis of its impact on model analysis in Appendix G.4 in our submission as follows. "As the augment ratio initially increases from 0 to 1.0, we observe improvements across performance metrics. However, when the augment ratio is increased beyond 1.0, these improvements become marginal or even negative. This indicates that excessive enhancement of the graph structure can increase learning difficulty. The increasing disparity between the enhanced and original graph topologies may also negatively impact performance. This study reveals that a reasonable augment ratio ε benefits the model by improving its sensitivity to bandwidth constraints. However, excessively high ε values provide only slight improvements or can even degrade performance." We also haved included the guidence of selection in revised manuscript, "Generally, setting ε = 1.0 or a value close to it provides a balanced tradeoff between performance enhancement and model robustness."
>
> [a] Jure Zbontar, et al. Barlow Twins: Self-Supervised Learning via Redundancy Reduction. ICML, 2021.
>
> [b] Yihao Xue, et al. Investigating Why Contrastive Learning Benefits Robustness against Label Noise. ICML, 2022.
>
> [c] Ting Chen, et al. A Simple Framework for Contrastive Learning of Visual Representations. ICML, 2020.
>
> [d] Zheye Deng, et al. GOLD: A Global and Local-aware Denoising Framework for Commonsense Knowledge Graph Noise Detection. EMNLP, 2023
>
> [e] Xuheng Cai, et al. LightGCL: Simple Yet Effective Graph Contrastive Learning for Recommendation. ICLR, 2023.

---

> ### Author Response · Authors · 2024-11-24
> **Author Response to W3-4**
>
> > **W3**: The integration of virtual and physical networks into a heterogeneous graph with numerous cross-graph links can lead to information redundancy and noise. Noise from irrelevant links can hinder the model's ability to learn meaningful representations.
>
> Thank you for your feedback. VNE  inherently involves embedding the cross-graph status between the virtual and physical networks, as the relationships between the two graphs play a crucial role in determining feasible and optimal solutions. To effectively capture these cross-graph relationships, we employ a heterogeneous graph modeling approach [a], which offers significant advantages over independently processing each graph with separate GNNs. Without cross-graph modeling, crucial relational information between the virtual and physical networks would be lost, limiting the model’s ability to learn representations that account for these dependencies.
>
> In our design, the addition of cross-graph links is not arbitrary but carefully constructed to reflect explicit semantic relationships between the nodes in the two graphs. Specifically, we introduce two types of heterogeneous links, each with clear semantics: one type connects virtual nodes to their potential physical hosts, and the other captures the constraints related to path connectivity between physical nodes. These explicitly defined links ensure that the model focuses on meaningful interactions, avoiding the introduction of random or noisy connections that could obscure learning. Additionally, the number of cross-graph links introduced is the same as the number of physical nodes, $N_p$, ensuring that the graph remains manageable in size and does not suffer from excessive redundancy.
>
> Heterogeneous graph modeling has been widely adopted in other domains with multiple interacting graphs, such as entity alignment in knowledge graphs [b], graph matching in computer vision [c], and optimizing Mixed-Integer Linear Programs [d], all of which demonstrate the effectiveness of this approach in learning meaningful cross-graph representations. Similarly, in our framework, the heterogeneous graph not only captures the nuanced relationships between the virtual and physical networks but also enhances the semantic richness of the learned embeddings, ultimately improving the performance of the VNE task.
>
> [a] Chuxu Zhang, et al. Heterogeneous Graph Neural Network. KDD, 2019
>
> [b] Jia-Chen Gu, et al. RHGN: Relation-gated Heterogeneous Graph Network for Entity Alignment in Knowledge Graphs. ACL, 2023.
>
> [c] Runzhong Wang, et al. Neural Graph Matching Network: Learning Lawler's Quadratic Assignment Problem with Extension to Hypergraph and Multiple-graph Matching. TPAMI, 2022
>
> [d] Ziang Chen, et al. On Representing Mixed-Integer Linear Programs by Graph Neural Networks. ICLR, 2023.
>
> > **W4**: The experiments are mainly conducted on simulated environments and limited network topologies (e.g., GEANT and BRAIN). This may not adequately demonstrate the model's performance.
>
> Thank you for your feedback. We conducted experiments on simulated environments following most existing studies in this direction. In the submission, we conducted the experiments in both simulated and real-world topologies. Specifically, we evaluated CONAL on WX100, WX500, GEANT, and BRAIN, covering a wide range of scalability and network densities:
>
> | Network | Number of Nodes | Number of Links | Network Density |
> |---------|-----------------|-----------------|-----------------|
> | WX100   | 100             | 500             | 0.05            |
> | WX500   | 500             | 13,000          | 0.1042          |
> | GEANT   | 40              | 64              | 0.0821          |
> | BRAIN   | 161             | 166             | 0.0129          |
>
> Additionally, we further consider the evaluation in various network conditions, such as varying arrival rates of VN requests and dynamic request distribution. To our best knowledge, this comprehensive evaluation setup is among the most thorough in the literature [e,f,g,h]. The chosen topologies effectively demonstrate CONAL's scalability, adaptability, and efficiency across different networking contexts.
>
> [e] Sheng Wu, et al. AI-Empowered Virtual Network Embedding:A Comprehensive Survey. IEEE Communications Surveys & Tutorials, 2024.
>
> [f] Song Yang, et al. Recent Advances of Resource Allocation in Network Function Virtualization. TPDS, 2021.
>
> [g] Haoyu Geng, et al. GAL-VNE: Solving the VNE Problem with Global Reinforcement Learning and Local One-Shot Neural Prediction. KDD, 2023.
>
> [h] More works in our submission's reference

---

> ### Author Response · Authors · 2024-11-24
> **Author Response to W5 & Q2**
>
> > **W5**: Given the rapid development of NFV, some relevant literatures are missing to be discussed, e.g., NFVdeep: Adaptive Online Service Function Chain Deployment with Deep Reinforcement Learning, iwqos’19; Adaptive VNF Scaling and Flow Routing with Proactive Demand Prediction, infocom’18; FlexNFV: Flexible Network Service Chaining with Dynamic Scaling, network’19; Joint Optimization of Chain Placement and Request Scheduling for Network Function Virtualization, icdcs’17, etc.
>
> Thank you for your suggestion. In the revised manuscript, we have expanded the Related Work section to include additional relevant literature, such as NFVdeep, Adaptive VNF Scaling, FlexNFV, JointOptimization, and more. The added discussions are as follows:
> - Lines 814-815 on Page 16: Resource management is a critical research direction in NFV, including tasks such as Scaling (Fei et al., 2018; 2020) and scheduling (Zhang et al., 2017). Among these, VNE plays a key role in resource allocation.
> - Lines 822-823 on Page 16: such as node ranking strategies (Zhang et al., 2018; Gong et al., 2014; Fan et al., 2023) (Jin et al., 2020)
> - Lines 825-826 on Page 16: many RL-based VNE algorithms have been proposed (Haeri & Trajkovi´c, 2017; Wang et al., 2021; Zhang et al., 2022; He et al., 2023a; Zhang et al., 2023b; Geng et al., 2023) (Xiao et al., 2019).
> These updates are highlighted in blue for clarity in the revised manuscript.
>
> > **Q2**: The paper does not provide a robust method for detecting infeasible instances, nor does it implement any controls or mitigations for the growth of λ. This oversight means that, when faced with infeasible instances, the model may fail to operate correctly.
>
> Thank you for your feedback. Detecting infeasible instances is computationally infeasible for NP-hard problems like VNE. Instead, our ARB approach dynamically adjusts reachability budgets, effectively mitigating the impact of unsolvable instances without explicit detection. This method is proposed to prevent λ from diverging, ensuring training stability and maintaining policy performance across various scenarios.
> We hope that these responses address your concerns and clarify the contributions and robustness of our method. Your feedback has been invaluable in improving the quality of our work, and we thank you once again for your thoughtful review.
>
> Again, we thank the reviewer for your in-depth suggestions for improving our submission. We hope the responses address your concerns. Thank you for your time and consideration.

---

> ### Author Response · Authors · 2024-12-02
> **Looking Forward to Further Discussion**
>
> Dear Reviewer S9pS,
>
> We sincerely appreciate your dedicated time and effort in reviewing our submission and providing thoughtful feedback. We greatly value your insights and hope our responses have adequately addressed your concerns.
>
> As the discussion deadline approaches, we would like to kindly invite you to share any additional feedback or questions you may have. We are more than happy to provide further details or clarifications.
>
> Thank you once again for your thoughtful comments. We look forward to hearing any further thoughts you might have.
>
> Best regards,
>
> The authors

---

### Author Response · Authors · 2024-11-28
**Global Clarification on Impacts**

Dear Reviewers,

Please allow us to further articulate our contributions to the ML/AI field, beyond the VNE problem.

Our work focuses on advancing machine learning for combinatorial optimization (ML4CO), a research area that has garnered significant attention in the ML community, as discussed in Related Work of our submission and highlighted in work [a,b]. While most existing studies in ML4CO primarily address classical problems such as Traveling Salesperson Problem (TSP) [c,d], Vehicle Routing Problem (VRP) [e,f], Job Shop Scheduling Problem (JSSP) [g,h], and VNE [i,j], they often overlook the complex constraints. However, many real-world applications are modeled as combinatorial optimization (CO) problems with complex constraints that are critical for real-world applicability. Effectively learning and managing these constraints in ML4CO remains a significant and challenging direction that has yet to be actively explored.

In this paper, we aim to push the boundaries of ML4CO by addressing a highly challenging constrained CO problem, i.e., VNE. VNE is characterized by hard and intricate constraints such as cross-graph resource allocation and bandwidth-constrained path routing. Distinct from most prior works, our focus lies in addressing the significant complexity introduced by these constraints and their practical implications. Below, we review our key contributions:

- *Revealing the Impact of Unsolvable Instances*: Through experimental observation and theoretical proof, we highlight the negative impact of unsolvable instances, which are inevitable in practical environments. Our analysis shows that such instances hinder the training process and policy performance, an issue largely overlooked in existing ML4CO research.
- *Innovative RL Optimization*: To stabilize training in the presence of unsolvable solutions, we propose a novel adaptive reachability budget. This innovation prevents divergence, ensures robust convergence in constrained scenarios, and is easily generalizable to other constrained CO problems.
- *Rethinking CMDP Modeling*: While existing works often model CO problems directly as MDPs or CMDPs, they simply stop when strict constraints are violated. we address this gap by introducing violation-tolerant CMDP modeling. This enables the complete exploration and precise evaluation of solution space, thus improving performance.
- *Well-designed Graph Representation*: Capturing complex constraints, particularly bandwidth-constrained path routing, is an unexplored and challenging area in GNN and ML4CO research. To address this, we design novel graph augmentation methods and leverage contrastive learning (CL) to improve bandwidth awareness and constraint representation.

Our work addresses the unique challenges posed by highly constrained CO problems, a less explored area in ML4CO, specifically in VNE. By revealing critical insights, and rethinking CMDP modeling, RL optimization, and GNN representation, we provide a pathway for applying ML to solve more complex and constrained CO problems across diverse domains. We believe our approach extends the frontiers of M4CO by introducing new paradigms in modeling, optimization, and representation, which are also broadly applicable to other constrained CO Problems beyond VNE.

We hope this response clarifies the significance of our work and its potential to advance the ML4CO community beyond VNE research.  We are committed to addressing any further questions or concerns you may have.

Reference
> [a] Yoshua Bengio, et al. Machine Learning for Combinatorial Optimization: a Methodological Tour d'Horizon. EJOR, 2020.
>
> [b] [Awesome Machine Learning for Combinatorial Optimization Resources](https://github.com/Thinklab-SJTU/awesome-ml4co)
>
> [c] Yifan Xia, et al. Position: Rethinking Post-Hoc Search-Based Neural Approaches for Solving Large-Scale Traveling Salesman Problems. ICML, 2024
>
> [d] Yimeng Min, et al. Unsupervised Learning for Solving the Travelling Salesman Problem. NeurIPS, 2023.
>
> [e] Qingchun Hou, et al. Generalize Learned Heuristics to Solve Large-scale Vehicle Routing Problems in Real-time. ICLR, 2023.
>
> [f] Jianan Zhou, et al. Towards Omni-generalizable Neural Methods for Vehicle Routing Problems ICML, 2023.
>
> [g] David W Zhang, et al. Robust Scheduling with GFlowNets. ICLR, 2023.
>
> [h] Wonseok Jeon, et al. Neural DAG Scheduling via One-Shot Priority Sampling. ICLR, 2023.
>
> [i] Tianfu Wang, et al. FlagVNE: A Flexible and Generalizable Reinforcement Learning Framework for Network Resource Allocation. IJCAI, 2024.
>
> [j] Haoyu Geng, et al. GAL-VNE: Solving the VNE Problem with Global Reinforcement Learning and Local One-Shot Neural Prediction. KDD, 2023.

---

### Meta-Review · Area_Chair_EM2s · 2024-12-20

**Metareview:**

The paper presents an RL algorithm that solves a combinatorial problem motivated from computer networking, namely, Virtual Network Embedding (VNE). The authors model this as a violation-tolerant Constrained Markov Decision Process. The authors also propose a constraint-aware graph representation method to efficiently learn cross-graph relations and constrained path connectivity in VNE.

From a practical standpoint, this problem and fast ML solutions to it seem to be of interest to the networking community. Reviewers however raised concerns whether studying this particular combinatorial-optimization problem and improving performance over competitors constitutes an algorithmic contribution of interest to the ICLR community. Some reviewers questioned the technical novelty; I would add to that one would expect that, at this point, an ML solution to combinatorial optimization problems would be evaluated more than one class of problems. Another recurring issue among reviewers was that there is no guarantee that constraints are satisfied.

**Additional Comments On Reviewer Discussion:**

Several reviewers remained concerned about the size of problems solved, the use of synthetic data, the purported latency for computation that would limit application to all but small examples and, most importantly, to the fact that solutions inevitably lead to violations of constraints.

---

### Decision · Program_Chairs · 2025-01-22

Reject